# C-C chemokine receptor 4 deficiency exacerbates early atherosclerosis in mice

Toru Tanaka[1†‡], Naoto Sasaki[1,2*†], Aga Krisnanda[1], Hilman Zulkifli Amin[1§, #, ¶], Ken Ito[1], Sayo Horibe[1], Kazuhiko Matsuo[3], Ken-ichi Hirata[2**], Takashi Nakayama[3], Yoshiyuki Rikitake[1]

[1]Laboratory of Medical Pharmaceutics, Kobe Pharmaceutical University, Kobe, Japan; [2]Division of Cardiovascular Medicine, Department of Internal Medicine, Kobe University Graduate School of Medicine, Kobe, Japan; [3]Division of Chemotherapy, Faculty of Pharmacy, Kindai University, Higashi-osaka, Japan

*For correspondence:
n-sasaki@kobepharma-u.ac.jp

†These authors contributed equally to this work

Present address: ‡Laboratory for inflammation and regeneration, College of Pharmaceutical Sciences, Ritsumeikan University, Shiga, Japan; §Integrated Clinical Education Center, Kobe University Hospital, Kobe, Japan; #Department of Cardiovascular Medicine, National Cerebral and Cardiovascular Center, Suita, Japan; ¶Faculty of Medicine, Universitas Indonesia, Jakarta, Indonesia; **Department of Cardiology, Kakogawa Central City Hospital, Kakogawa, Japan

Competing interest: The authors declare that no competing interests exist.

## eLife Assessment

This **valuable** study provides in vivo evidence that CCR4 regulates the early inflammatory response during atherosclerotic plaque formation. The authors propose that altered T-cell response plays a role in this process, shedding light on mechanisms that may be of interest to medical biologists, biochemists, cell biologists, and immunologists. The work is currently considered **incomplete** pending textual changes and the inclusion of proper controls.

## Abstract

Chronic inflammation via dysregulation of T cell immune responses is critically involved in the pathogenesis of atherosclerotic cardiovascular disease. Improving the balance between proinflammatory T cells and anti-inflammatory regulatory T cells (Tregs) may be an attractive approach for treating atherosclerosis. Although C-C chemokine receptor 4 (CCR4) has been shown to mediate the recruitment of T cells to inflamed tissues, its role in atherosclerosis is unclear. Here, we show that genetic deletion of CCR4 in hypercholesterolemic mice accelerates the development of early atherosclerotic lesions characterized by an inflammatory plaque phenotype. This was associated with the augmentation of proinflammatory T helper type 1 (Th1) cell responses in peripheral lymphoid tissues, para-aortic lymph nodes, and atherosclerotic aorta. Mechanistically, CCR4 deficiency in Tregs impaired their suppressive function and tended to inhibit their migration to the atherosclerotic aorta, and subsequently augmented Th1 cell-mediated immune responses through defective regulation of dendritic cell function, which accelerated aortic inflammation and atherosclerotic lesion development. Thus, we revealed a previously unrecognized role for CCR4 in controlling the early stage of atherosclerosis via Treg-dependent regulation of proinflammatory T cell responses. Our data suggest that CCR4 is an important negative regulator of atherosclerosis.

## Introduction

Severe cardiovascular diseases, including ischemic heart disease and stroke, occur in patients with atherosclerosis and are major causes of mortality worldwide. Despite state-of-the-art intensive treatment, patients at high risk of atherosclerotic diseases have considerable residual risk, which could be partly related to vascular inflammatory responses (*Weber et al., 2023*). Notably, recent clinical trials have provided evidence that anti-inflammatory therapies could be potentially novel approaches for preventing cardiovascular diseases in patients with a past disease history, although overall mortality has not improved (*Ridker et al., 2017*; *Tardif et al., 2019*).

Chronic aortic inflammation via T cell-mediated immune dysregulation has been shown to be critically involved in the development of atherosclerosis (*Roy et al., 2022*). Single-cell proteomic and transcriptomic approaches have revealed that activated and differentiated T cells are the major immune cells infiltrating human carotid artery plaques (*Fernandez et al., 2019*), indicating the involvement of T cell-mediated immunoinflammatory responses in atherosclerotic plaque development in humans. A recent clinical trial provided evidence for the increased risk of atherosclerotic cardiovascular disease events in cancer patients treated with immune checkpoint inhibitors (*Drobni et al., 2020*), confirming the critical role of effector T cell (Teff) immune responses in the development of human atherosclerotic disease. Teffs, including T helper type 1 (Th1) cells, play a proatherogenic role by producing proinflammatory cytokines, including interferon (IFN)-γ (*Gupta et al., 1997*; *Buono et al., 2005*). A recent translational study demonstrated that anti-CXC-motif-chemokine receptor 3 autoantibodies, which reflect Th1 cell-mediated responses, can be a novel biomarker and an inflammatory risk factor for cardiovascular morbidity and mortality beyond traditional risk factors, indicating the possible proatherogenic role of Th1 cells in humans (*Müller et al., 2023*). On the other hand, several subsets of regulatory T cells (Tregs), including forkhead box P3 (Foxp3)-expressing Tregs, play an anti-atherogenic role by suppressing Teff-mediated immunoinflammatory responses or reducing plasma atherogenic lipoprotein levels (*Ait-Oufella et al., 2006*; *Sasaki et al., 2009*; *Klingenberg et al., 2013*). A number of approaches to dampening proatherogenic Th1 cell responses or augmenting atheroprotective Treg responses have been reported to be effective for the treatment and prevention of atherosclerosis in experimental mouse models (*Tanaka et al., 2021*). Despite accumulating evidence for the critical role of the Th1 cell/Treg balance in atherosclerosis, how the balance of these T cell populations is regulated in lymphoid tissues and atherosclerotic lesions to control atherosclerosis is unclear.

Various immune cells, including T cells and monocytes, migrate to the aorta via interactions between chemokines and their specific chemokine receptors and are critically involved in the process of atherosclerosis (*Noels et al., 2019*). However, few reports have identified a chemokine system that prevents atherosclerosis by improving the Th1 cell/Treg balance in lymphoid tissues and atherosclerotic lesions. C-C chemokine receptor 4 (CCR4) is expressed on several T cell subsets, including T helper type 2 (Th2) cells, T helper type 17 (Th17) cells, skin-homing T cells, and Tregs, but not on proatherogenic Th1 cells (*Griffith et al., 2014*). CCR4 is a highly specific receptor for two CC chemokine ligands CCL17 (thymus- and activation-regulated chemokine) and CCL22 (macrophage-derived chemokine) (*Yoshie, 2021*), which were shown to be expressed in mesenteric lymph nodes (LNs) in a mouse model of inflammatory bowel disease (*Yuan et al., 2007*) and lung tissues affected by allergic airway inflammation (*Faustino et al., 2013*). These chemokines play a role in guiding CCR4+ T cells to the inflammatory sites with abundant expression of these chemokines (*Campbell et al., 1999*; *Montane et al., 2011*). Previous experimental studies have suggested that the CCL17/CCL22–CCR4 axes protect against inflammatory autoimmune diseases partly by promoting Treg accumulation in target tissues (*Yuan et al., 2007*; *Faustino et al., 2013*). The expression of CCL17 and CCL22 was also observed in human and mouse atherosclerotic lesions (*Greaves et al., 2001*; *Weber et al., 2011*; *Kimura et al., 2015*). Although the disruption of the CCL17/CCL22–CCR4 axes did not affect the development of advanced atherosclerotic lesions or the proportion of Tregs in peripheral lymphoid tissues in atherosclerosis-prone mice fed a high-cholesterol diet (*Weber et al., 2011*; *Döring et al., 2024*), its effect on the early stage of atherosclerosis and Treg responses has not been investigated.

Here, we investigated the role of CCR4 in the development of early atherosclerotic lesions and the underlying mechanisms in CCR4-deficient (*Ccr4*-/-) mice on an apolipoprotein E-deficient (*Apoe*-/-) background fed a standard chow diet, with a particular focus on CD4+ T cell immune responses.

## Results

### CCR4 is predominantly expressed on CD4+Foxp3+ Tregs, and the ligands CCL17 and CCL22 are expressed in peripheral LNs and atherosclerotic lesions

To evaluate the expression levels of CCR4 on CD4+ T cells under normocholesterolemic or hypercholesterolemic conditions, we performed flow cytometric analysis of peripheral LNs, spleen, and para-aortic LNs from wild-type, *Apoe*-/-, and *Ccr4*-/-*Apoe*-/- mice. Notably, CCR4 expression was expressed on approximately 15–25% of CD4+Foxp3+ Tregs from wild-type or *Apoe*-/- mice (*Appendix 1—figure*

1A–C), while CD4+Foxp3- non-Tregs from these mice expressed CCR4 at markedly lower levels, suggesting that CCR4 is predominantly expressed on CD4+Foxp3+ Tregs and that hypercholesterolemia does not affect CCR4 expression. We also examined CCR4 expression on CD4+Foxp3+ Tregs in the atherosclerotic aorta of Apoe-/- mice and consistently found that CCR4 expression in CD4+Foxp3+ Tregs was much higher than that in CD4+Foxp3- non-Tregs (Appendix 1—figure 1D).

As CCL17 and CCL22, known as specific ligands for CCR4, are highly expressed by dendritic cells (DCs) in LNs (Alferink et al., 2003; Tang and Cyster, 1999), we examined their expression in the peripheral LNs of wild-type, Apoe-/-, and Ccr4-/-Apoe-/- mice by immunohistochemistry. As expected, clear expression of these chemokines was observed in the peripheral LNs of wild-type, Apoe-/-, and Ccr4-/-Apoe-/- mice (Appendix 2—figure 1). We also examined the expression of CCL17 and CCL22 in the aorta of these mice by immunohistochemistry. CCL17 expression was detected in the atherosclerotic lesions of Apoe-/- and Ccr4-/-Apoe-/- mice, and some lesional MOMA-2+ macrophages expressed this chemokine (Appendix 3—figure 1A). However, CCL17 expression was not detected in the aorta of wild-type mice without atherosclerotic plaques (Appendix 3—figure 1A). Another ligand CCL22 was modestly expressed in atherosclerotic lesions and partially colocalized with lesional MOMA-2+ macrophages in Apoe-/- or Ccr4-/-Apoe-/- mice, while its expression was not detected in the aorta of wild-type mice (Appendix 3—figure 1B).

Together, these findings suggest that Tregs may migrate to peripheral LNs and atherosclerotic lesions partly via the CCL17/CCL22–CCR4 axes under hypercholesterolemia.

## CCR4 deficiency accelerates the development of early atherosclerotic lesions characterized by an inflammatory plaque phenotype

To investigate the effect of CCR4 deficiency on the development of early atherosclerosis, we analyzed the atherosclerotic lesions of 18-week-old Apoe-/- and Ccr4-/-Apoe-/- mice fed a standard chow diet. Ccr4-/-Apoe-/- mice developed normally without any spontaneous inflammatory disease. Notably, compared with Apoe-/- mice, Ccr4-/-Apoe-/- mice exhibited a significant increase in atherosclerotic lesion size in the aortic sinus (aortic sinus mean plaque area: control Apoe-/- mice: $1.46\pm0.50\times10^5$ µm² versus Ccr4-/-Apoe-/-: $2.04\pm0.82 \times 10^5$ µm², p<0.05; Figure 1A). In parallel with the cross-sectional studies, we performed en face analysis of thoracoabdominal aortas, revealing no significant difference in aortic plaque burden between the two groups (Figure 1B). There were no significant differences in body weight or plasma lipid profile between the two groups (Table 1).

To determine the effect of CCR4 deficiency on plaque components, we performed immunohistochemical studies of atherosclerotic lesions in the aortic sinus. Notably, compared with those of Apoe-/- mice, the atherosclerotic lesions of Ccr4-/-Apoe-/- mice showed a 20% increase in macrophage accumulation (Figure 1C) and a marked 42% increase in CD4+ T cell infiltration (Figure 1D). In addition, we performed immunohistochemical analysis of Foxp3+ Tregs in atherosclerotic lesions using an anti-Foxp3 antibody. However, few Foxp3+ Tregs were found within the plaques of Apoe-/- or Ccr4-/-Apoe-/- mice (data not shown). The proportion of collagen in the aortic sinus plaques of Ccr4-/-Apoe-/- mice was significantly lower than that in the aortic sinus plaques of Apoe-/- mice (Figure 1E). These findings on atherosclerotic plaque size and components collectively suggest that CCR4 plays a role in preventing the development of early atherosclerotic lesions and inducing a less inflammatory plaque phenotype. To further evaluate aortic immunoinflammatory responses, we analyzed the mRNA expression of pro- and anti-inflammatory cytokines and transcription factors specific for Tregs or helper T cell subsets in atherosclerotic aorta by quantitative reverse transcription PCR. The mRNA expression of proinflammatory cytokines (Il1b and Il6), Th1-related Tbx21, and Th17-related Rorc was markedly upregulated in the aorta of Ccr4-/-Apoe-/- mice, indicating augmented proatherogenic immune responses in the atherosclerotic aorta (Figure 1F). The mRNA expression of the Treg-specific transcription factor Foxp3 was undetectable.

Collectively, these data suggest that CCR4 deficiency promotes the accumulation of inflammatory cells and proinflammatory immune responses in the aorta, leading to augmented development of early atherosclerotic lesions in the aortic root.

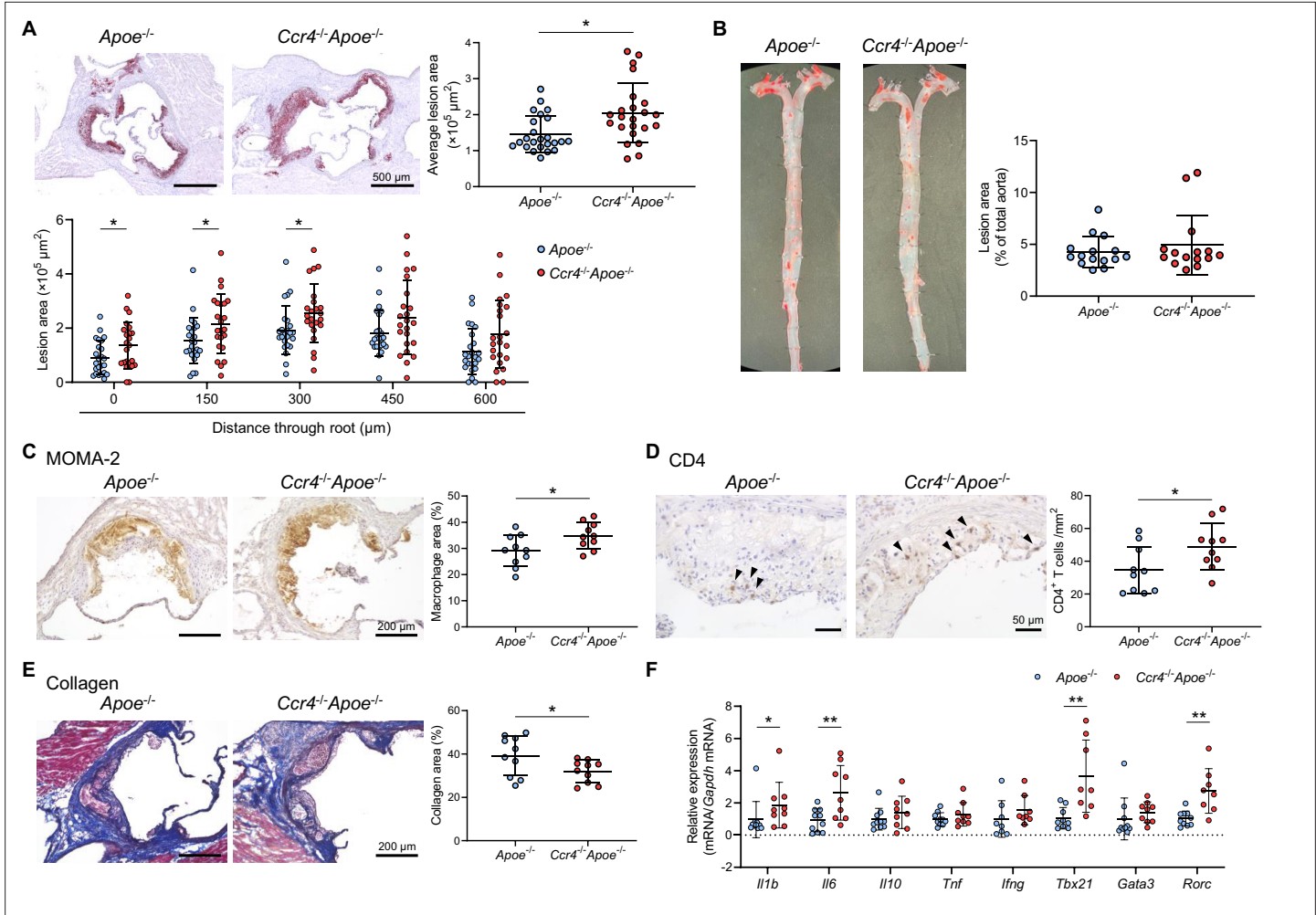

**Figure 1.** C-C chemokine receptor 4 (CCR4) deficiency accelerates the development of early atherosclerotic lesions characterized by an inflammatory plaque phenotype. (**A**) Representative photomicrographs of Oil Red O staining and quantitative analysis of atherosclerotic lesion area at five different levels and the average area in the aortic sinus of 18-week-old apolipoprotein E-deficient (*Apoe⁻ᐟ⁻*) mice (n=24) or CCR4-deficient mice on an *Apoe⁻ᐟ⁻* background (*Ccr4⁻ᐟ⁻Apoe⁻ᐟ⁻*; n=23). (**B**) Representative photomicrographs of Oil Red O staining and quantitative analysis of atherosclerotic lesion area in the aorta of 18-week-old *Apoe⁻ᐟ⁻* (n=15) or *Ccr4⁻ᐟ⁻Apoe⁻ᐟ⁻* mice (n=15). (**C–E**) Representative sections and quantitative analyses of MOMA-2⁺ macrophages (**C**), CD4⁺ T cells (**D**), and collagen (**E**) in the aortic sinus. Arrowheads indicate the CD4⁺ T cells. n=10 per group. (**F**) mRNA expression of pro- or anti-inflammatory cytokines and helper T cell-associated transcription factors in aorta. The expression levels of the target genes were normalized so that the mean values in *Apoe⁻ᐟ⁻* mice were set to 1. n=8–10 per group. Eighteen-week-old *Apoe⁻ᐟ⁻* or *Ccr4⁻ᐟ⁻Apoe⁻ᐟ⁻* mice were used for all experiments. Black bars represent 50, 200, or 500 µm as described. Data points represent individual animals. Horizontal bars represent means. Error bars indicate s.d. *p<0.05, **p<0.01; Mann–Whitney *U*-test: (**A**) and (**F**) *l1b*; two-tailed Student's *t*-test: (**C–F**) *Il6*, *Tbx21*, and *Rorc*.

The online version of this article includes the following source data for figure 1:

**Source data 1.** Raw numerical values for *Figure 1* plots.

**Table 1.** Body weight and plasma lipid profile in 18-week-old mice.

|  | *Apoe⁻ᐟ⁻* | *Ccr4⁻ᐟ⁻Apoe⁻ᐟ⁻* |
|---|---|---|
| Body weight (g) | 31.76±1.94 (n=27) | 31.69±2.24 (n=27) |
| Total cholesterol (mg/dL) | 526.5±146.3 (n=10) | 520.4±150.6 (n=10) |
| High-density lipoprotein-cholesterol (mg/dL) | 24.90±6.49 (n=10) | 19.80±5.87 (n=10) |
| Triglycerides (mg/dL) | 89.70±21.07 (n=10) | 86.40±39.58 (n=10) |

# CCR4 deficiency augments Teff immune responses in peripheral lymphoid tissues

We examined the mechanisms by which CCR4 deficiency accelerates early atherosclerosis by focusing on changes in systemic T cell responses, including those involving CD4+Foxp3+ Tregs and CD4+Foxp3- non-Tregs, in peripheral lymphoid tissues. The frequencies of CD4+Foxp3+ Tregs and CD4+CD44high CD62Llow effector memory T cells were significantly higher in the spleen of 8- or 18-week-old *Ccr4-/-Apoe-/-* mice than in those of age-matched *Apoe-/-* mice (**Figure 2A and B**). A similar tendency was observed for the peripheral LNs of these mice, although there was no difference in the frequency of CD4+CD44high CD62Llow effector memory T cells between 8-week-old *Apoe-/-* and *Ccr4-/-Apoe-/-* mice (**Figure 2A and B**). The absolute numbers of CD4+Foxp3+ Tregs and CD4+CD44high-CD62Llow effector memory T cells were significantly higher in the peripheral LNs of 8- or 18-week-old *Ccr4-/-Apoe-/-* mice than those of age-matched *Apoe-/-* mice (**Figure 2A and B**). The absolute number of CD4+CD44high CD62Llow effector memory T cells was also higher in the spleen of 18-week-old *Ccr4-/-Apoe-/-* mice than that of age-matched *Apoe-/-* mice (**Figure 2B**). Similar results were obtained for normocholesterolemic wild-type mice (**Figure 2—figure supplement 1A and B**), suggesting that the expansion of Tregs and effector memory T cells in *Ccr4-/-Apoe-/-* mice is independent of hypercholesterolemia. We evaluated the proliferative capacity of CD4+Foxp3+ Tregs and CD4+Foxp3- non-Tregs by analyzing Ki-67 expression using flow cytometry and found that the proportions of Ki-67-positive Tregs and non-Tregs were markedly higher in the peripheral LNs and spleen of *Ccr4-/-Apoe-/-* mice than in those of *Apoe-/-* mice (**Figure 2C and D**). To determine the effect of CCR4 deficiency on T cell development in the thymus, we performed flow cytometric analysis of thymocytes from 4-week-old *Apoe-/-* or *Ccr4-/-Apoe-/-* mice and found no difference in the development of thymic T cells between the two groups (**Figure 2—figure supplement 2A**). In line with a previous report in normocholesterolemic mice (**Lee et al., 2005**), there was no difference in the frequency of thymic CD4+Foxp3+ Tregs between the two groups (**Figure 2—figure supplement 2B**). These data indicate that the increased numbers of CD4+Foxp3+ Tregs and CD4+Foxp3- non-Tregs in peripheral LNs are due to their enhanced proliferative capacity but not to their promoted development in the thymus. We also analyzed other immune cells in the spleen of *Apoe-/-* and *Ccr4-/-Apoe-/-* mice by flow cytometry and found no major differences in the proportions or activation-associated molecule expression between the two groups, except for CD86 expression on DCs which was upregulated in *Ccr4-/-Apoe-/-* mice (**Figure 2—figure supplement 3**).

To determine the activation and function of CCR4-deficient CD4+Foxp3+ Tregs, we investigated the expression of their activation- and function-associated molecules in the peripheral LNs of 8- or 18-week-old *Apoe-/-* or *Ccr4-/-Apoe-/-* mice by flow cytometry. Notably, CCR4 deficiency had no major effect on the expression of cytotoxic T lymphocyte-associated antigen-4 (CTLA-4) or CD103 in CD4+Foxp3+ Tregs (**Figure 2E**). We also analyzed the mRNA expression of *Ccr4*, activation- or function-associated molecules (*Foxp3*, *Ctla4*, *Cd103*, *Tnfrsf18*, *Il10*, and *Tgfb*), and major chemokine receptors (*Ccr5*, *Ccr6*, *Ccr7*, and *Ccr8*) in splenic Tregs by quantitative reverse transcription PCR. The *Ccr4* mRNA expression in Tregs from *Ccr4-/-Apoe-/-* mice was less than the detectable levels (**Figure 2—figure supplement 4A**). In line with the data on peripheral LNs, there were no significant differences in the mRNA expression of these molecules between the two groups (**Figure 2F**, **Figure 2—figure supplement 4A**).

In contrast to the results on CD4+Foxp3+ Tregs, the expression of the activation marker CTLA-4 in peripheral LN CD4+Foxp3- non-Tregs was higher in *Ccr4-/-Apoe-/-* mice than in *Apoe-/-* mice, although the expression of another activation marker CD103 was not altered by CCR4 deficiency (**Figure 2G**). We analyzed the mRNA expression of *Ccr4*, activation-associated molecules (*Ctla4*, *Cd44*, *Cd69*, and *Cd103*), and representative chemokine receptors (*Ccr1*, *Ccr5*, *Ccr6*, *Ccr7*, *Ccr8*, *Cxcr3*, and *Cx3cr1*) in splenic non-Tregs by quantitative reverse transcription PCR. The *Ccr4* mRNA expression in non-Tregs from *Ccr4-/-Apoe-/-* mice was less than the detectable levels (**Figure 2—figure supplement 4B**). We found a marked increase in the mRNA expression of activation-associated molecules (*Ctla4*, *Cd44*, and *Cd103*) in splenic non-Tregs from *Ccr4-/-Apoe-/-* mice (**Figure 2H**). We also observed upregulated mRNA expression of various chemokine receptors, including Th1-related *Ccr1*, *Cxcr3*, and *Cx3cr1*, Th2-related *Ccr8*, and Th17-related *Ccr6* in splenic non-Tregs from *Ccr4-/-Apoe-/-* mice (**Figure 2—figure supplement 4B**), indicating that CCR4 deficiency may promote the activation and expansion of helper T cells and facilitate their migration to atherosclerotic lesions. There were no significant

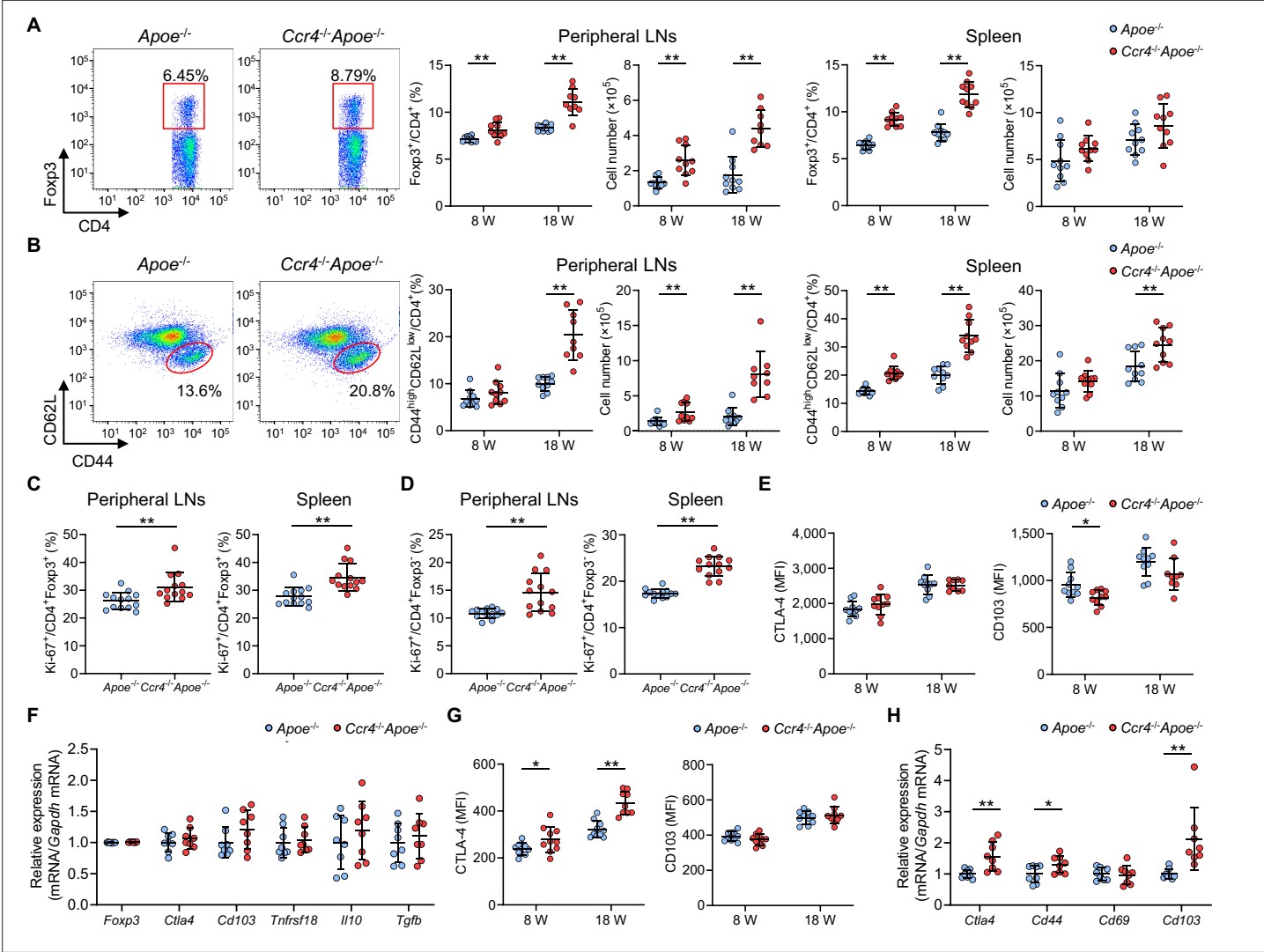

**Figure 2.** C-C chemokine receptor 4 (CCR4) deficiency augments effector T cell immune responses in peripheral lymphoid tissues. (**A, B**) Representative flow cytometric analysis of CD4[+] forkhead box P3 (Foxp3)[+] regulatory T cells (Tregs) (**A**) and CD4[+]CD44[high]CD62L[low] effector memory T cells (**B**) in the spleen of 8-week-old apolipoprotein E-deficient (*Apoe*[-/-]) mice or CCR4-deficient mice on an *Apoe*[-/-] background (*Ccr4*[-/-]*Apoe*[-/-]). The graphs represent the total numbers and proportions of CD4[+]Foxp3[+] Tregs (**A**) and CD4[+]CD44[high]CD62L[low] effector memory T cells (**B**) in the peripheral lymph nodes (LNs) and spleen of 8- or 18- week-old *Apoe*[-/-] or *Ccr4*[-/-]*Apoe*[-/-] mice. n=9–10 per group. (**C, D**) The graphs represent the proportions of Ki-67-positive cells among CD4[+]Foxp3[+] Tregs (**C**) and CD4[+]Foxp3[-] non-Tregs (**D**) in the peripheral LNs and spleen of 8-week-old *Apoe*[-/-] or *Ccr4*[-/-]*Apoe*[-/-] mice, as assessed by flow cytometry. n=13 per group. (**E**) Expression levels of activation-associated molecules cytotoxic T lymphocyte-associated antigen-4 (CTLA-4) and CD103 were analyzed by gating on CD4[+]Foxp3[+] Tregs in the peripheral LNs of 8- or 18-week-old *Apoe*[-/-] or *Ccr4*[-/-]*Apoe*[-/-] mice. n=9–10 per group. (**F**) mRNA expression of Treg-associated markers in splenic Tregs from 8-week-old *Apoe*[-/-] or *Ccr4*[-/-]*Apoe*[-/-] mice. n=8 per group. (**G**) Expression levels of activation-associated molecules CTLA-4 and CD103 were analyzed by gating on CD4[+]Foxp3[-] non-Tregs in the peripheral LNs of 8- or 18-week-old *Apoe*[-/-] or *Ccr4*[-/-]*Apoe*[-/-] mice. n=9–10 per group. (**H**) mRNA expression of activation-associated molecules in splenic non-Tregs from 8-week-old *Apoe*[-/-] or *Ccr4*[-/-]*Apoe*[-/-] mice. n=8 per group. The expression levels of the target genes were normalized so that the mean values in *Apoe*[-/-] mice were set to 1 (**F, H**). Data points represent individual animals. Horizontal bars represent means. Error bars indicate s.d. *p<0.05, **p<0.01; Mann–Whitney *U*-test: (**A**) second (8w) from the left, (**B**) second (8w) and third (8w) from the left, (**C**, left, and **H**) *Cd44* and *Cd103*; two-tailed Student's *t*-test: (**A**) first, second (18w), and third from the left, (**B**) first, second (18w), third (18w), and fourth from the left, (**C**, right, **D**, **E**, **G**, and **H**) *Ctla4*. MFI, mean fluorescence intensity.

The online version of this article includes the following source data and figure supplement(s) for figure 2:

**Source data 1.** Raw numerical values for *Figure 2* plots.

**Figure supplement 1.** CCR4 deficiency expands peripheral Tregs and effector memory T cells in the peripheral lymphoid tissues of normocholesterolemic mice.

**Figure supplement 1—source data 1.** Raw numerical values for *Figure 2—figure supplement 1* plots.

*Figure 2 continued on next page*

*Figure 2 continued*

**Figure supplement 2.** CCR4 deficiency does not affect T cell development in the thymus.

**Figure supplement 2—source data 1.** Raw numerical values for *Figure 2—figure supplement 2* plots.

**Figure supplement 3.** CCR4 deficiency has a minor effect on the proportions of other immune cells in the spleen.

**Figure supplement 3—source data 1.** Raw numerical valuse for *Figure 2—figure supplement 3* plots.

**Figure supplement 4.** The effect of CCR4 deficiency on the expression of various chemokine receptors in splenic Tregs and non-Tregs.

**Figure supplement 4—source data 1.** Raw numerical values for *Figure 2—figure supplement 4* plots.

differences in the mRNA expression of other chemokine receptors (*Ccr5* and *Ccr7*) in splenic non-Tregs between the two groups (*Figure 2—figure supplement 4B*).

Considering the accelerated early atherosclerosis observed in *Ccr4*[-/-]*Apoe*[-/-] mice, these results indicate that augmented Teff immune responses may affect atherosclerosis more strongly than the increase in Tregs in peripheral lymphoid tissues.

## CCR4 deficiency promotes proinflammatory CD4[+] T cell immune responses in peripheral lymphoid tissues

To determine whether CCR4 deficiency affects CD4[+] T cell immune responses and polarization, we examined cytokine secretion from CD4[+] T cells by intracellular cytokine staining. The fraction of IFN-γ-producing Th1 cells in the peripheral LNs was significantly higher in *Ccr4*[-/-]*Apoe*[-/-] mice than in *Apoe*[-/-] mice, while there were no differences in the fractions of other CD4[+] T cell subsets including interleukin (IL)-4-producing Th2 cells, IL-10-producing CD4[+] T cells, and IL-17-producing Th17 cells between the two groups (*Figure 3A*). In line with this, the fraction of splenic Th1 cells was higher in *Ccr4*[-/-]*Apoe*[-/-] mice than in *Apoe*[-/-] mice, although the proportion of IL-17-producing Th17 cells was also higher in *Ccr4*[-/-]*Apoe*[-/-] mice than in *Apoe*[-/-] mice (*Figure 3B*).

We semiquantitatively analyzed the production of various cytokines or chemokines by splenic CD4[+] T cells stimulated with plate-bound anti-CD3 and anti-CD28 antibodies using a cytokine array kit. Compared with those from *Apoe*[-/-] mice, splenic CD4[+] T cells from *Ccr4*[-/-]*Apoe*[-/-] mice secreted more Th1-related cytokine IFN-γ, Th2-related cytokine IL-13, Th17-related cytokine IL-17, and various inflammation-related cytokines and chemokines (*Figure 3C*). ELISA analysis confirmed that the IFN-γ and IL-17 levels in the cell supernatants of *Ccr4*[-/-]*Apoe*[-/-] mice were much higher than those in the cell supernatants of *Apoe*[-/-] mice (*Figure 3D*). Although the cytokine levels of Th2-related cytokine IL-4 and Treg-related anti-inflammatory cytokine IL-10 were below the detectable levels by cytokine array analysis, ELISA analysis revealed higher IL-4 and IL-10 production in splenic CD4[+] T cells from *Ccr4*[-/-]*Apoe*[-/-] mice than in those from *Apoe*[-/-] mice (*Figure 3D*), which may be compensatory immune responses to the proinflammatory T cell responses caused by CCR4 deficiency.

## CCR4 deficiency promotes Th1 cell responses in para-aortic LNs and atherosclerotic aorta

Next, we investigated the mechanisms by which CCR4 deficiency accelerates early atherosclerosis by focusing on local immune responses in para-aortic LNs and atherosclerotic aorta. Consistent with the peripheral LN data, the frequency and number of CD4[+]Foxp3[+] Tregs were significantly higher in the para-aortic LNs of *Ccr4*[-/-]*Apoe*[-/-] mice than in those of *Apoe*[-/-] mice (*Figure 4A*). The number of CD4[+]CD44[high]CD62L[low] effector memory T cells was significantly higher in the para-aortic LNs of *Ccr4*[-/-]*Apoe*[-/-] mice than in those of *Apoe*[-/-] mice, while their frequency did not differ between the two groups (*Figure 4B*).

In line with the data on peripheral lymphoid tissues, there were no significant differences in the expression of activation- or function-associated molecules or chemokine receptors in para-aortic LN Tregs between the two groups (*Figure 4C*, *Figure 4—figure supplements 1 and 2A*). Notably, the expression of *Cd103* and Th1-related *Tbx21* was upregulated in para-aortic LN non-Tregs from *Ccr4*[-/-]*Apoe*[-/-] mice, while the expression of other activation- or helper T cell-associated molecules or chemokine receptors was unaltered (*Figure 4D*, *Figure 4—figure supplement 2B*). Flow cytometric analysis of helper T cell subsets in para-aortic LNs revealed that the fractions of IFN-γ-producing Th1 cells, IL-4-producing Th2 cells, and IL-17-producing Th17 cells were significantly higher in *Ccr4*[-/-]*Apoe*[-/-] mice

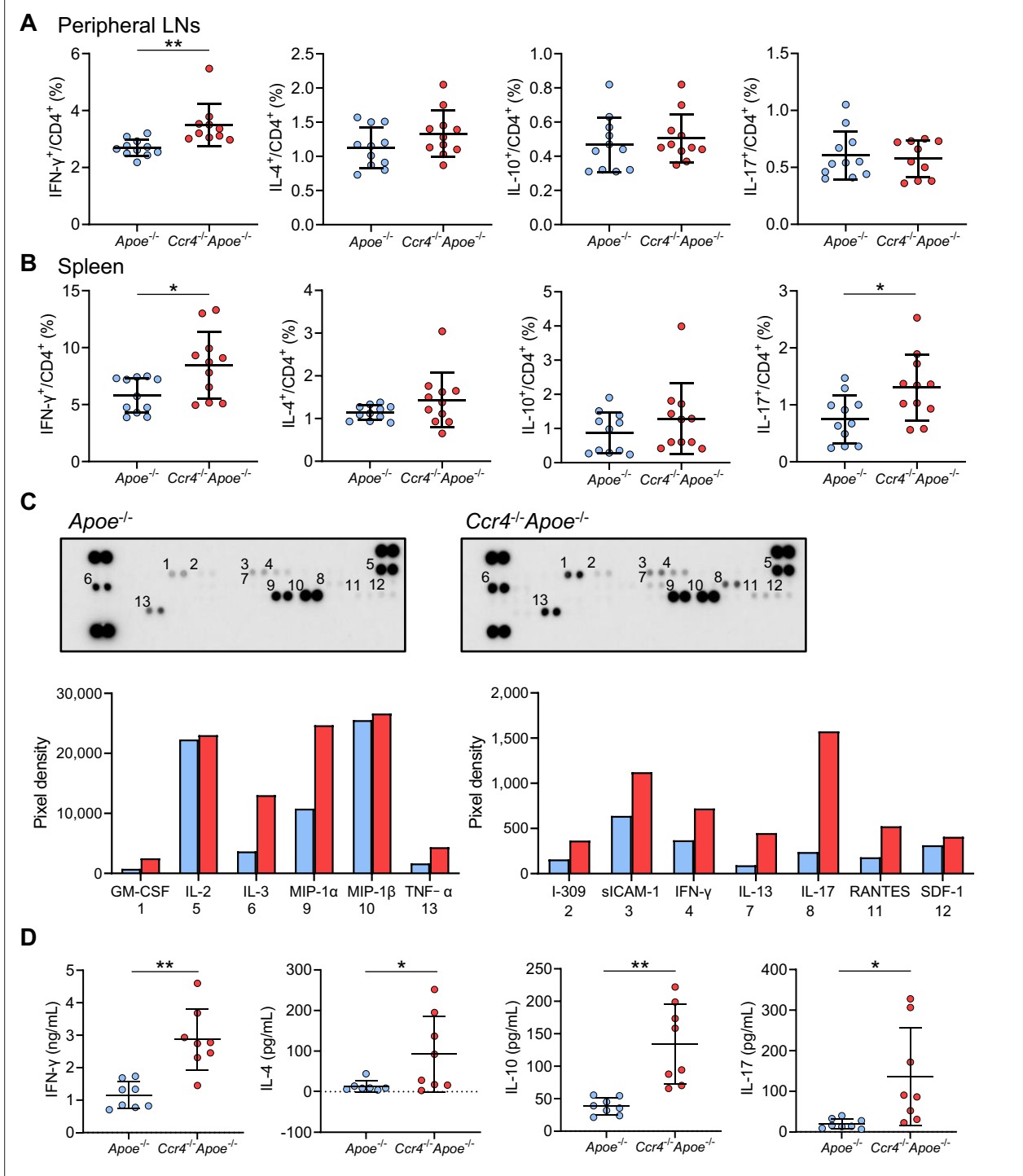

**Figure 3.** C-C chemokine receptor 4 (CCR4) deficiency promotes proinflammatory CD4+ T cell immune responses in peripheral lymphoid tissues. (**A, B**) The graphs represent the frequencies of interferon (IFN)-γ+, interleukin (IL)–4+, IL-10+, and IL-17+ CD4+ T cells in the peripheral lymph nodes (LNs) (**A**) and spleen (**B**) of 8-week-old apolipoprotein E-deficient (*Apoe-/-*) mice or CCR4-deficient mice on an *Apoe-/-* background (*Ccr4-/-Apoe-/-*). n=10–11 per group. (**C, D**) Purified splenic CD4+ T cells from 8-week-old *Apoe-/-* or *Ccr4-/-Apoe-/-* mice were stimulated with plate-bound anti-CD3 and soluble anti-CD28 antibodies in vitro. The levels of various cytokines and chemokines in pooled cell supernatants from eight mice in each group were determined semiquantitatively by a cytokine array kit (**C**). Data are representative of two independent experiments. Cytokine concentrations in the cell supernatants were measured by ELISA (**D**). n=8 per group. Data points represent individual animals. Horizontal bars represent means. Error bars indicate s.d. *p<0.05, **p<0.01; Mann–Whitney *U*-test: (**A**, **B**) first from the left, and (**D**) second from the left; two-tailed Student's *t*-test: (**B**) fourth from the left and (**D**) first, third, and fourth from the left.

*Figure 3 continued on next page*

*Figure 3 continued*

The online version of this article includes the following source data for figure 3:

**Source data 1.** Raw numerical values for *Figure 3* plots.

than in *Apoe*[-/-] mice, suggesting augmented proinflammatory CD4[+] T cell immune responses by CCR4 deficiency (*Figure 4E*).

We explored the infiltration of T-box expressed in T cells (T-bet)-expressing Th1 cells, GATA3-expressing Th2 cells, and retinoic acid-related orphan receptor gamma t (RORγt)-expressing Th17 cells in atherosclerotic aorta by flow cytometry. Strikingly, the frequency of aortic T-bet-expressing Th1 cells was markedly higher in *Ccr4*[-/-]*Apoe*[-/-] mice than in *Apoe*[-/-] mice, while the proportions of Th2 and Th17 subsets did not differ between the two groups (*Figure 4F–H*). We next analyzed the accumulation of CD4[+]Foxp3[+] Tregs in the atherosclerotic aorta by flow cytometry and found no difference in the proportion of CD4[+]Foxp3[+] Tregs between the two groups (*Figure 4I*). Importantly, we found a marked increase in the ratio of T-bet-expressing Th1 cells to CD4[+]Foxp3[+] Tregs (Th1 cell/Treg ratio) in the atherosclerotic aorta of *Ccr4*[-/-]*Apoe*[-/-] mice (*Figure 4J*), indicating Th1-skewed immune responses in the atherosclerotic lesions of *Ccr4*[-/-]*Apoe*[-/-] mice.

Collectively, these results suggest that CCR4 deficiency promotes the selective accumulation of proatherogenic Th1 cells in atherosclerotic aorta and the accumulation of various helper T cell subsets including Th1 cells in para-aortic LNs and shifts the Th1 cell/Treg balance toward Th1 cell responses in atherosclerotic aorta, leading to exacerbated aortic inflammation and early atherosclerosis.

## CCR4 expression on Tregs regulates Th1 cell responses and may mediate Treg migration to the atherosclerotic aorta

Given that CCR4 plays a crucial role in attenuating immunoinflammatory responses in autoimmune or allergic disease via the modulation of Treg function (*Yuan et al., 2007*; *Faustino et al., 2013*), we speculated that despite the expansion and unaltered expression of activation- or function-associated molecules in Tregs, these cells might be dysfunctional. To determine whether CCR4 deficiency affects the suppressive function of Tregs in hypercholesterolemia, we performed an in vitro suppression assay. Interestingly, CCR4 deficiency significantly impaired the suppressive function of Tregs isolated from hypercholesterolemic mice (*Figure 5A*), indicating that CCR4 expression on Tregs may be important for the regulation of proinflammatory immune responses and the development of early atherosclerosis.

Interaction with DCs is well-known as one of the core suppressive mechanisms by which Tregs control excessive immune responses. A previous report demonstrated that the cell–cell contacts between Tregs and CCL22-deficient DCs are impaired (*Rapp et al., 2019*). Tregs limit the CD80/CD86–CD28-dependent activation of T cells through CTLA-4-dependent downregulation of CD80 and CD86 expression on DCs (*Sakaguchi et al., 2020*), which may contribute to the reduction in atherosclerosis (*Matsumoto et al., 2016*). Therefore, we examined the interactions between CCR4-intact or CCR4-deficient Tregs and DCs by a coculture experiment (*Figure 5B*). As expected, the upregulation of CD80 and CD86 expression on DCs mediated by conventional T cells was markedly suppressed by coculture with Tregs from *Apoe*[-/-] or *Ccr4*[-/-]*Apoe*[-/-] mice (*Figure 5C*). Notably, the suppressive effect of CCR4-deficient Tregs was significantly attenuated compared with that of CCR4-intact Tregs (*Figure 5C*). To assess the involvement of CCR4 expression on Tregs in regulating the production of proatherogenic IFN-γ by conventional T cells, we stimulated conventional T cells with an anti-CD3 antibody and DCs in the presence or absence of CCR4-intact or CCR4-deficient Tregs and analyzed IFN-γ production by ELISA (*Figure 5B*). IFN-γ production by conventional T cells was markedly suppressed by coculture with CCR4-intact Tregs, while it was not significantly affected by coculture with CCR4-deficient Tregs (*Figure 5D*). Importantly, there was a marked difference in the ability to suppress the production of IFN-γ from conventional T cells between CCR4-intact Tregs and CCR4-deficient Tregs (*Figure 5D*). These results suggest that the augmented Th1 cell responses in the peripheral lymphoid tissues of *Ccr4*[-/-]*Apoe*[-/-] mice are partly due to impaired Treg-dependent regulation of DC function.

As described above, CD4[+]Foxp3[+] Treg accumulation in the atherosclerotic aorta of *Ccr4*[-/-]*Apoe*[-/-] mice was not altered despite the marked expansion of CD4[+]Foxp3[+] Tregs in peripheral lymphoid tissues. Based on these findings and previous reports showing the impaired migratory capacity of

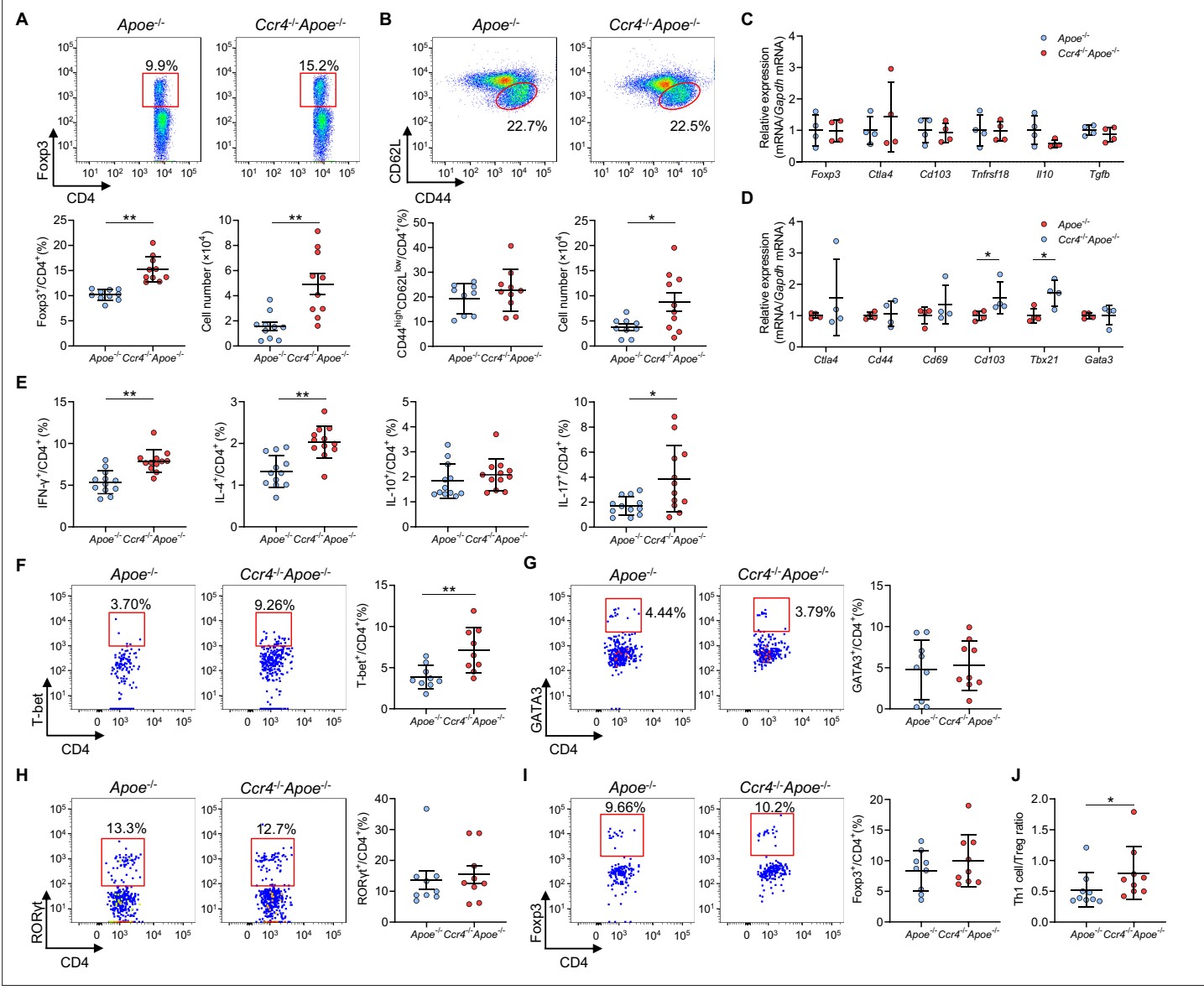

**Figure 4.** C-C chemokine receptor 4 (CCR4) deficiency promotes T helper type 1 (Th1) cell responses in para-aortic lymph nodes (LNs) and atherosclerotic aorta. (**A, B**) Representative flow cytometric analysis of CD4+ forkhead box P3 (Foxp3)+ regulatory T cells (Tregs) (**A**) and CD4+CD44highCD62Llow effector memory T cells (**B**) in para-aortic LNs. The graphs represent the total numbers and proportions of CD4+Foxp3+ Tregs (**A**) and CD4+CD44highCD62Llow effector memory T cells (**B**) in para-aortic LNs. n=9–10 per group. (**C, D**) mRNA expression of Treg-associated markers in Tregs (**C**) and mRNA expression of activation or helper T cell-associated molecules in non-Tregs (**D**) in para-aortic LNs. The expression levels of the target genes were normalized so that the mean values in apolipoprotein E-deficient (*Apoe*-/-) mice were set to 1. Tregs or non-Tregs purified from pooled para-aortic LNs of 9–10 mice were analyzed as a sample. n=4 per group. (**E**) The graphs represent the frequencies of interferon (IFN)-γ+, interleukin (IL)–4+, IL-10+, and IL-17+ CD4+ T cells in para-aortic LNs. n=12 per group. (**F–H**) Representative flow cytometric analysis of T-box expressed in T cells (T-bet) (**F**), GATA3 (**G**), and retinoic acid-related orphan receptor gamma t (RORγt) (**H**) expression in aortic CD3+CD4+CD45+ T cells. The graphs represent the frequencies of T-bet+ (**F**), GATA3+ (**G**), and RORγt+ (**H**) cells among aortic CD3+CD4+CD45+ T cells. n=9 per group. (**I**) Representative flow cytometric analysis of Foxp3 expression in aortic CD3+CD4+CD45+ T cells. The graph represents the frequency of Foxp3+ Tregs among aortic CD3+CD4+CD45+ T cells. n=9 per group. (**J**) The graph represents the ratio of CD4+T-bet+ Th1 cells to CD4+Foxp3+ Tregs (Th1 cell/Treg ratio). n=9 per group. Pooled aortic lymphoid cells from two mice were analyzed as a sample. Eighteen-week-old *Apoe*-/- or CCR4-deficient mice on an *Apoe*-/- background (*Ccr4*-/-*Apoe*-/-) were used for all experiments. Data points represent individual animals (**A**, **B**, and **E**) or individual pooled samples (**C**, **D**, **F–J**). Horizontal bars represent means. Error bars indicate s.d. *p<0.05, **p<0.01; Mann–Whitney U-test: (**D**) *Cd103* and (**J**); two-tailed Student's t-test: (**A, B, D**) *Tbx21*, (**E, F**).

The online version of this article includes the following source data and figure supplement(s) for figure 4:

**Source data 1.** Raw numerical values for *Figure 4* plots.

*Figure 4 continued on next page*

*Figure 4 continued*

**Figure supplement 1.** CCR4 deficiency does not affect the expression of activation- or function-associated molecules in Tregs in para-aortic lymph nodes (LNs).

**Figure supplement 1—source data 1.** Raw numerical values for *Figure 4—figure supplement 1* plots.

**Figure supplement 2.** The effect of CCR4 deficiency on the expression of various chemokine receptors in Tregs and non-Tregs in para-aortic lymph nodes (LNs).

**Figure supplement 2—source data 1.** Raw numerical values for *Figure 4—figure supplement 2* plots.

CCR4-deficient Tregs to inflammatory tissues (*Yuan et al., 2007*; *Faustino et al., 2013*), we hypothesized that CCR4-deficient Tregs might have less capacity to migrate to atherosclerotic lesions. To address this issue, we performed a Treg transfer experiment using *Apoe*-/- or *Ccr4*-/-*Apoe*-/- mice on a Kaede-Tg background (*Tomura et al., 2008*), which provided us with the opportunity to faithfully track transferred Tregs by monitoring the fluorescent Kaede protein (*Figure 5E*). There was no difference in the migratory capacity of CCR4-intact or CCR4-deficient Kaede-expressing Tregs to the peripheral LNs, spleen, or para-aortic LNs of recipient *Apoe*-/- mice (*Figure 5F–H*). To promote the accumulation of T cells in atherosclerotic aorta, we fed recipient *Apoe*-/- mice with a high-cholesterol diet and analyzed the migration of transferred Tregs in the aorta. Interestingly, we found a trend toward reduction in the proportion of CCR4-deficient Kaede-expressing Tregs in the aorta of recipient *Apoe*-/- mice (*Figure 5I*), suggesting that CCR4-deficient Tregs may have a reduced ability to migrate to the atherosclerotic aorta, but not to the peripheral lymphoid tissues, under hypercholesterolemia.

Taken together, these data demonstrate that CCR4 expression on Tregs plays a critical role in regulating Th1 cell responses in lymphoid tissues and may mediate Treg migration to the atherosclerotic aorta under hypercholesterolemia, which may cooperatively contribute to the reduction in early atherosclerosis by efficiently mitigating aortic inflammatory immune responses.

## CCR4 expression on Tregs is critical for limiting aortic inflammation and the development of atherosclerosis

To provide direct evidence for the critical role of CCR4 expression in Tregs in reducing early atherosclerosis, we injected *Apoe*-/- mice with saline or Tregs from *Apoe*-/- or *Ccr4*-/-*Apoe*-/- mice and analyzed the aortic root atherosclerotic lesions of recipient *Apoe*-/- mice (*Figure 6A*). Although there were no significant differences in the aortic sinus mean plaque area among the three groups (*Figure 6—figure supplement 1*), detailed analysis of the aortic root plaques at five different levels revealed significantly greater lesions in *Apoe*-/- mice that were administered CCR4-deficient Tregs than in *Apoe*-/- mice that were administered CCR4-intact Tregs (*Figure 6B*), suggesting that the impaired CCR4-deficient Treg function is involved in the acceleration of atherosclerosis in *Ccr4*-/-*Apoe*-/- mice. There were no significant differences in body weight or plasma lipid profile among the three groups (*Table 2*). We further aimed to confirm these findings by performing an additional experiment in which recipient *Ccr4*-/-*Apoe*-/- mice were administered Tregs from *Apoe*-/- or *Ccr4*-/-*Apoe*-/- mice or PBS and atherosclerotic lesions were analyzed (*Figure 6—figure supplement 2A*). There were no major differences in body weight or plasma lipid profile among the three groups (*Table 3*). The anti-atherogenic effect of CCR4 expression in Tregs was not observed in these mice (*Figure 6—figure supplement 2B*). This could possibly be explained by the dysfunction of Tregs under the enhanced inflammatory conditions in *Ccr4*-/-*Apoe*-/- mice.

To determine the effect of Treg-specific CCR4 deficiency on plaque components, we performed immunohistochemical studies of atherosclerotic lesions in the aortic sinus. Compared with those of saline-injected *Apoe*-/- mice, the atherosclerotic lesions of *Apoe*-/- mice injected with CCR4-intact Tregs showed markedly reduced accumulation of macrophages (*Figure 6C*) and CD4+ T cells (*Figure 6D*), whereas there were no differences in intraplaque accumulation of these inflammatory cells between saline-injected *Apoe*-/- mice and *Apoe*-/- mice injected with CCR4-deficient Tregs (*Figure 6C and D*). Notably, macrophage accumulation in the aortic sinus atherosclerotic lesions was markedly higher in *Apoe*-/- mice injected with CCR4-deficient Tregs than in *Apoe*-/- mice injected with CCR4-intact Tregs (*Figure 6C*). Collagen content in atherosclerotic lesions was significantly higher in *Apoe*-/- mice injected with CCR4-intact Tregs than in saline-injected *Apoe*-/- mice, while no difference in collagen

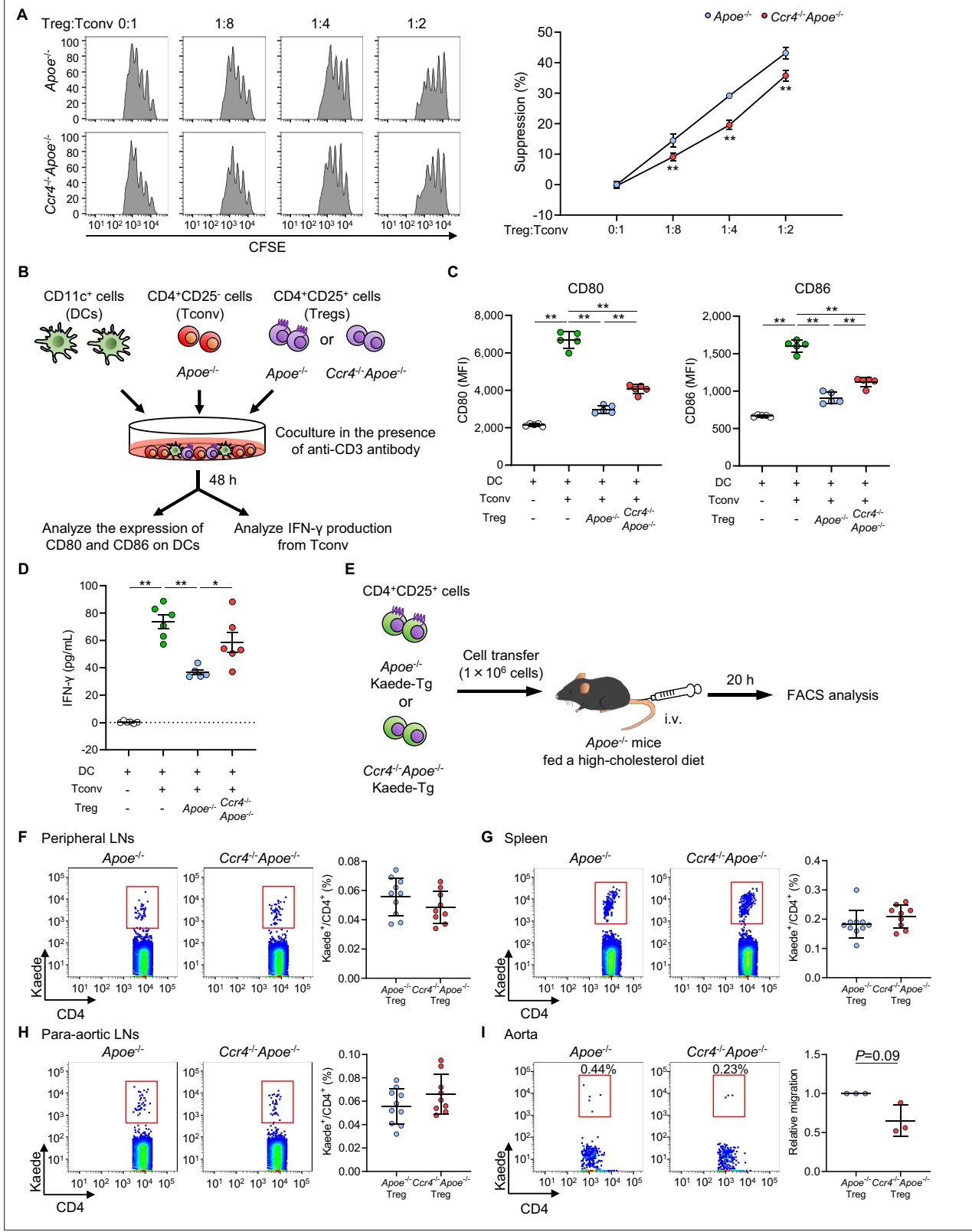

**Figure 5.** C-C chemokine receptor 4 (CCR4) expression on regulatory T cells (Tregs) regulates T helper type 1 cell responses and mediates Treg migration to the aorta. (**A**) The suppressive function of Tregs was assessed by evaluating the proliferation of carboxyfluorescein diacetate succinimidyl ester (CFSE)-labeled conventional T cells (Tconv) cocultured with Tregs from apolipoprotein E-deficient (*Apoe⁻/⁻*) mice or CCR4-deficient mice on an *Apoe⁻/⁻* background (*Ccr4⁻/⁻Apoe⁻/⁻*). Data are presented as the results of triplicate wells and are representative of two independent experiments. Data

*Figure 5 continued on next page*

*Figure 5 continued*

are expressed as the mean ± s.d. (**B, C**) CD80 and CD86 expression in live splenic dendritic cells (DCs) after 2 days of coculture with Tconv from *Apoe*⁻/⁻ mice, or a mixture of Tregs from *Apoe*⁻/⁻ or *Ccr4*⁻/⁻*Apoe*⁻/⁻ mice and Tconv from *Apoe*⁻/⁻ mice in the presence of an anti-CD3 antibody. Data points represent the results of quintuplicate wells. Data are representative of two independent experiments. (**B, D**) Tconv from *Apoe*⁻/⁻ mice and DCs were cocultured with or without Tregs from *Apoe*⁻/⁻ or *Ccr4*⁻/⁻*Apoe*⁻/⁻ mice in the presence of an anti-CD3 antibody. Interferon (IFN)-γ concentrations in cell supernatants were measured by ELISA. Data points represent the results of sextuplicate wells. (**E**) Eighteen-week-old *Apoe*⁻/⁻ mice fed a high-cholesterol diet for 10 weeks received transfer of Tregs from *Apoe*⁻/⁻Kaede-Tg or *Ccr4*⁻/⁻*Apoe*⁻/⁻Kaede-Tg mice, and the accumulation of Kaede⁺ Tregs in the peripheral lymphoid tissues and aorta was analyzed by flow cytometry 20 hours later. (**F–I**) Representative flow cytometric analysis and the proportions of Kaede⁺ Tregs among CD4⁺ T cells in the peripheral lymph nodes (LNs) (**F**), spleen (**G**), para-aortic LNs (**H**), and aorta (**I**) of *Apoe*⁻/⁻ mice that received *Apoe*⁻/⁻Kaede⁺ Tregs or *Ccr4*⁻/⁻*Apoe*⁻/⁻Kaede⁺ Tregs. n=9–10 per group (**F–H**). Pooled aortic lymphoid cells from five mice in each group were used for analysis. The results are presented as the mean ±s.d. of three independent experiments (**I**). Data points represent individual animals (**F–H**) or individual pooled samples (**I**). Horizontal bars represent means. Error bars indicate s.d. *p<0.05, **p<0.01; one-way ANOVA followed by Tukey's multiple-comparisons test: (**C, D**); two-way ANOVA followed by Tukey's multiple-comparisons test: (**A**). p=0.09; one-sample *t*-test: (**I**). MFI, mean fluorescence intensity.

The online version of this article includes the following source data for figure 5:

**Source data 1.** Raw numerical values for *Figure 5* plots.

was observed between saline-injected *Apoe*⁻/⁻ mice and *Apoe*⁻/⁻ mice injected with CCR4-deficient Tregs (*Figure 6E*).

Overall, we provide evidence that CCR4 protects against early atherosclerosis partly by mediating Treg-dependent induction of a less inflammatory plaque phenotype.

## Discussion

In the present study, we demonstrated that genetic deletion of CCR4 in hypercholesterolemic *Apoe*⁻/⁻ mice accelerates the development of early atherosclerotic lesions in the aortic root that exhibit an inflammatory phenotype, associated with the augmentation of proatherogenic Th1 cell responses in the peripheral lymphoid tissues, para-aortic LNs, and atherosclerotic aorta. Furthermore, T cell-DC coculture and Treg transfer experiments revealed that CCR4 expression on Tregs regulates the development of early atherosclerosis by suppressing Th1 cell responses in lymphoid tissues and possibly by mediating Treg migration to the atherosclerotic aorta. Thus, we identified a novel role for the CCL17/CCL22–CCR4 axes in controlling early atherosclerosis via favorable modulation of the Th1 cell/Treg balance.

Emerging experimental and clinical data obtained by single-cell RNA sequencing and mass cytometry have clearly shown that CD4⁺ or CD8⁺ T cells are the dominant populations in human (*Fernandez et al., 2019*) and mouse (*Winkels et al., 2018*) atherosclerotic plaques. Recent experimental and clinical evidence has demonstrated that dysregulation of the balance of proatherogenic Th1 cells and anti-atherogenic Tregs has adverse effects on atherosclerotic disease (*Tanaka et al., 2021*). In the present study, we found that the formation of early atherosclerotic lesions in the aortic root was markedly accelerated by CCR4 deficiency, which was associated with the upregulation of various helper T cell immune responses in peripheral lymphoid tissues and augmented Th1 and Th17 cell-mediated responses in the atherosclerotic aorta. Given the proatherogenic nature of Th1 cells (*Gupta et al., 1997*; *Buono et al., 2005*), our data indicate that augmented Th1 cell responses in peripheral lymphoid tissues, especially in atherosclerotic lesions, substantially contributed to atherosclerotic lesion formation in *Ccr4*⁻/⁻*Apoe*⁻/⁻ mice. However, we cannot exclude the possibility that augmented immune responses of other helper T cell subsets might also contribute to accelerated atherosclerosis in *Ccr4*⁻/⁻*Apoe*⁻/⁻ mice.

As Th2 cells and some Th17 cells, but not Th1 cells, are known to express CCR4 (*Griffith et al., 2014*), the augmented Th1 cell responses observed in *Ccr4*⁻/⁻*Apoe*⁻/⁻ mice do not seem to be direct effects of CCR4 deficiency and may involve other mechanisms. Given our recent report showing the augmented immune responses related to Th1, Th2, and Th17 cells in CD4⁺Foxp3⁺ Treg-depleted atherosclerosis-prone mice (*Kasahara et al., 2022*), augmentation of these helper T cell responses may be caused by dysregulated CD4⁺Foxp3⁺ Treg responses in our *Ccr4*⁻/⁻*Apoe*⁻/⁻ mice. In line with experimental studies in mouse models of inflammatory diseases showing a pivotal role for CCR4 in mediating Treg migration to inflammatory tissues (*Yuan et al., 2007*; *Faustino et al., 2013*), we found that CCR4 expression on Tregs may be critical for their migration to the atherosclerotic aorta,

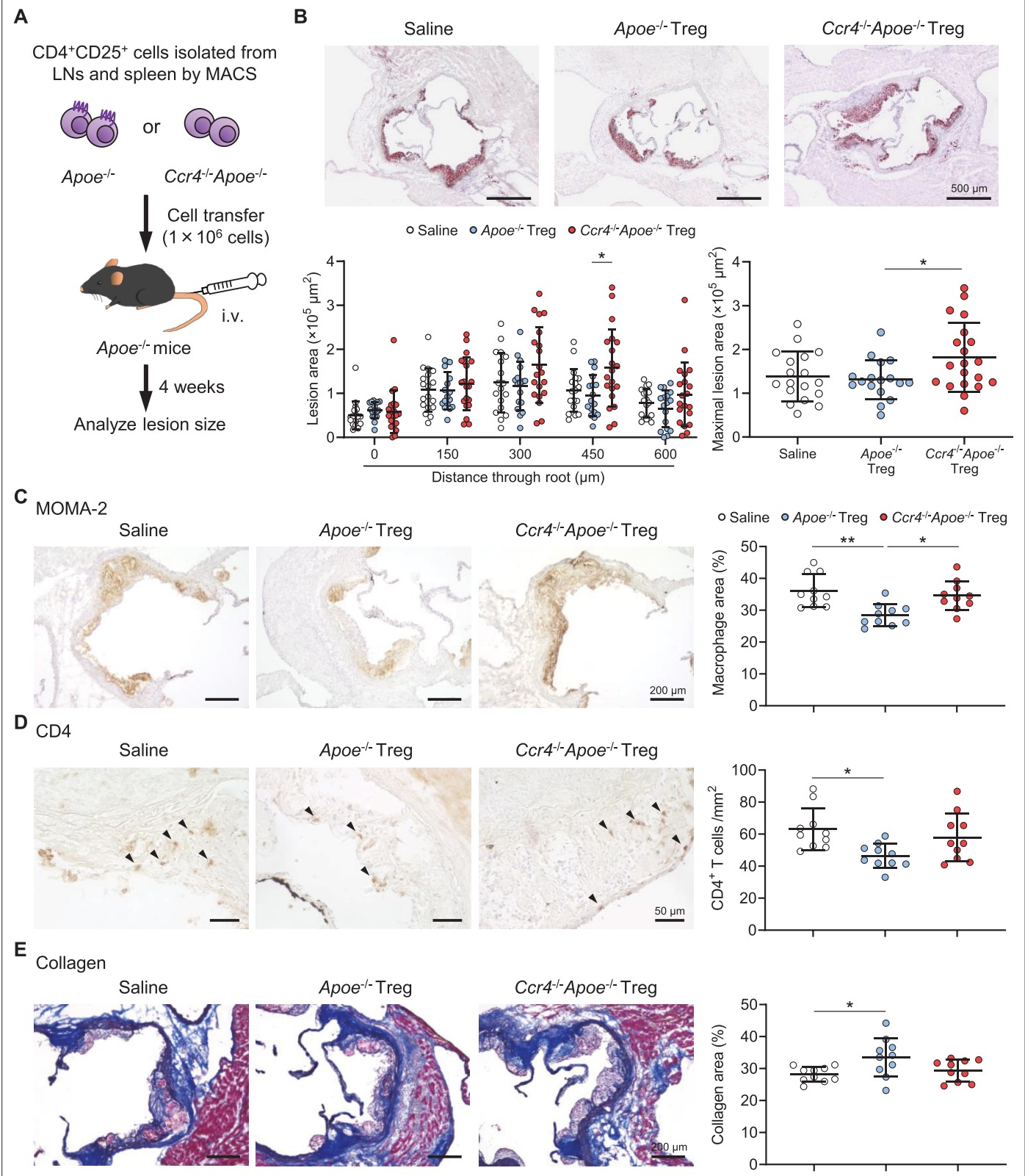

**Figure 6.** C-C chemokine receptor 4 (CCR4) expression on regulatory T cells (Tregs) is critical for limiting aortic inflammation and the development of early atherosclerosis. (**A**) Tregs purified from the peripheral lymph nodes (LNs) and spleen of apolipoprotein E-deficient (*Apoe*⁻/⁻) mice or CCR4-deficient mice on an *Apoe*⁻/⁻ background (*Ccr4*⁻/⁻*Apoe*⁻/⁻) were intravenously transferred into 12-week-old *Apoe*⁻/⁻ mice fed a standard chow diet, and atherosclerotic lesions were analyzed at 16 weeks of age. As a control without cell transfer, 12-week-old *Apoe*⁻/⁻ mice were intravenously injected with

*Figure 6 continued on next page*

*Figure 6 continued*

saline and atherosclerotic lesions were analyzed at 16 weeks of age. (**B**) Representative photomicrographs of Oil Red O staining and quantitative analysis of atherosclerotic lesion area at five different levels and maximal lesions in the aortic sinus of *Apoe*[-/-] mice injected with saline (n=19), *Apoe*[-/-] Tregs (n=17), or *Ccr4*[-/-]*Apoe*[-/-] Tregs (n=20). (**C–E**) Representative sections and quantitative analyses of MOMA-2[+] macrophages (**C**), CD4[+] T cells (**D**), and collagen (**E**) in the aortic sinus of *Apoe*[-/-] mice injected with saline, *Apoe*[-/-] Tregs, or *Ccr4*[-/-]*Apoe*[-/-] Tregs. Arrowheads indicate the CD4[+] T cells. n=10 per group. Black bars represent 50, 200, or 500 µm as described. Data points represent individual animals. Horizontal bars represent means. Error bars indicate s.d. *p<0.05, **p<0.01; one-way ANOVA followed by Tukey's multiple-comparisons test.

The online version of this article includes the following source data and figure supplement(s) for figure 6:

**Source data 1.** Raw numerical values for *Figure 6* plots.

**Figure supplement 1.** No significant difference in the mean atherosclerotic lesion area in the aortic sinus was observed among *Apoe*[-/-] mice injected with saline, *Apoe*[-/-] Tregs, or *Ccr4*[-/-]*Apoe*[-/-] Tregs.

**Figure supplement 1—source data 1.** Raw numerical values for *Figure 6—figure supplement 1* plots.

**Figure supplement 2.** Transfer of CCR4-intact Tregs does not affect the development of early atherosclerotic lesions in *Ccr4*[-/-]*Apoe*[-/-] mice.

**Figure supplement 2—source data 1.** Raw numerical values for *Figure 6—figure supplement 2* plots.

providing a possible mechanism for CCR4-dependent regulation of atherosclerosis. Another interesting finding is that CCR4 deficiency markedly upregulated Th1 cell responses in peripheral lymphoid tissues as well as in atherosclerotic lesions under hypercholesterolemia. This observation may not depend on the activation status of Tregs because there were no major differences in the expression of their activation- and function-associated molecules between *Apoe*[-/-] and *Ccr4*[-/-]*Apoe*[-/-] mice. Despite the lower expression of CD103 in the peripheral LN Tregs of 8-week-old *Ccr4*[-/-]*Apoe*[-/-] mice, there was no significant difference in its expression levels between 18-week-old *Apoe*[-/-] and *Ccr4*[-/-]*Apoe*[-/-] mice, indicating that reduced CD103 expression in *Ccr4*[-/-]*Apoe*[-/-] mice may not be a noteworthy change. T cell-DC coculture experiments clearly demonstrated that CCR4 expression on Tregs is critical for attenuating DC-dependent stimulation of Th1 cell responses by maintaining interactions between Tregs and DCs and Treg suppressive function, which is supported by the findings of a previous report showing a crucial role of the CCL22–CCR4 axis in the contact of Tregs with DCs and regulation of inflammatory responses (*Rapp et al., 2019*). In line with the results of in vitro experiments, we found the upregulation of CD86 expression on DCs in *Ccr4*[-/-]*Apoe*[-/-] mice, which may be derived from the impaired ability of CCR4-deficient Tregs to downregulate CD80 and CD86 expression on DCs. Based on these findings, we propose that CCR4 deficiency in Tregs interrupts their contact with DCs in lymphoid tissues and impairs their suppressive function, leading to augmented Th1 cell responses and accelerated atherosclerosis. We speculate that the increased frequency of Tregs in the peripheral lymphoid tissues of *Ccr4*[-/-]*Apoe*[-/-] mice may be a compensatory effect in response to augmented Th1 cell-mediated proinflammatory responses due to dysfunctional Tregs.

T cell costimulatory signals such as the CD28–CD80/CD86 (*Ait-Oufella et al., 2006*) and CD27–CD70 (*Winkels et al., 2017*) pathways protect against atherosclerosis by systemically shifting the Th1 cell/Treg balance toward Treg responses. Several approaches involving treatment with cytokines (*Dinh et al., 2012*; *Kasahara et al., 2014*), antibodies (*Sasaki et al., 2009*; *Kasahara et al., 2014*; *Steffens et al., 2006*), an active form of vitamin D$_3$ (*Takeda et al., 2010*), or ultraviolet B irradiation *Sasaki et al., 2017*; *Tanaka et al., 2024* have been reported to mitigate atherosclerosis via modulation of the Th1 cell/Treg balance in the peripheral lymphoid tissues and atherosclerotic lesions of atherosclerosis-prone mice. Despite these reports, the mechanisms that regulate the T cell balance in lymphoid tissues, particularly in atherosclerotic lesions, have not been fully elucidated. The chemokine system plays an important role in the differentiation and migration of various subsets of T cells

**Table 2.** Body weight and plasma lipid profile in *Apoe*[-/-] mice treated with saline, *Apoe*[-/-] Tregs, or *Ccr4*[-/-]*Apoe*[-/-] Tregs.

| | Saline | *Apoe*[-/-]Treg | *Ccr4*[-/-]*Apoe*[-/-]Treg |
|---|---|---|---|
| Body weight (g) | 27.82±2.82 (n=18) | 27.80±2.42 (n=20) | 27.72±2.59 (n=19) |
| Total cholesterol (mg/dL) | 505.3±114.0 (n=10) | 605.5±105.1 (n=10) | 607.3±76.35 (n=10) |
| High-density lipoprotein-cholesterol (mg/dL) | 15.00±2.87 (n=10) | 15.20±3.26 (n=10) | 13.80±4.83 (n=10) |
| Triglycerides (mg/dL) | 87.40±36.69 (n=10) | 92.70±27.15 (n=10) | 96.60±33.94 (n=10) |

**Table 3.** Body weight and plasma lipid profile in *Ccr4⁻/⁻Apoe⁻/⁻* mice treated with saline, *Apoe⁻/⁻* Tregs, or *Ccr4⁻/⁻Apoe⁻/⁻* Tregs.

| | Saline | *Apoe⁻/⁻* Treg | *Ccr4⁻/⁻Apoe⁻/⁻* Treg |
|---|---|---|---|
| Body weight (g) | 32.44±2.75 (n=12) | 30.69±3.55 (n=12) | 32.16±2.12 (n=12) |
| Total cholesterol (mg/dL) | 589.1±144.3 (n=10) | 616.1±105.6 (n=10) | 526.4±115.9 (n=10) |
| High-density lipoprotein-cholesterol (mg/dL) | 16.30±3.65 (n=10) | 13.80±3.19 (n=10) | 12.10±2.96* (n=10) |
| Triglycerides (mg/dL) | 83.0±57.56 (n=10) | 91.60±55.52 (n=10) | 72.70±13.99 (n=10) |

including Tregs and has been shown to be involved in the process of atherosclerosis (*Noels et al., 2019*). We revealed that CCR4 protects against early atherosclerosis by mitigating Th1 cell responses in lymphoid tissues and atherosclerotic lesions and possibly by mediating Treg migration to the aorta. This indicates a previously unrecognized role of the chemokine system in regulating the Th1 cell/Treg balance to limit atherosclerosis.

We observed the accelerated development of early atherosclerotic lesions in *Ccr4⁻/⁻Apoe⁻/⁻* mice fed a standard chow diet. However, previous studies showed that neither hematopoietic nor systemic CCR4 deficiency affected the development of advanced atherosclerotic lesions or Treg frequency in severely hypercholesterolemic mice (*Weber et al., 2011*; *Döring et al., 2024*). These findings suggest that the role of CCR4 may vary depending on the stages of atherosclerosis. Given that Treg responses affected by CCR4 deficiency in mice with early atherosclerotic lesions may be different from those in mice with advanced atherosclerotic lesions, we speculate that the difference in Treg responses may be responsible for the inconsistent findings on the role of CCR4 deficiency in atherosclerosis.

Our data show that interactions between CCR4 and CCL17/CCL22 may promote Treg-skewed responses in lymphoid tissues and atherosclerotic lesions. However, studies in hypercholesterolemic mice demonstrated that genetic deficiency of CCL17 ameliorated atherosclerotic lesion development by promoting Treg accumulation in lymphoid tissues and atherosclerotic lesions independently of the CCL17–CCR4 axis (*Weber et al., 2011*; *Döring et al., 2024*). This suggests that CCL17 restrains Treg homeostasis and accelerates atherosclerosis, although the role of the CCL17–CCR4 axis in the regulation of Treg homeostasis and atherosclerosis remains elusive. A study in mouse models of myocardial injury showed that CCL17 deficiency increased Treg recruitment to the heart and attenuated myocardial inflammation and injury, which may be attributed to the cancellation of competitive inhibition of CCL22-mediated Treg chemotaxis by CCL17 (*Feng et al., 2022*), suggesting that CCL17 may impair Treg homeostasis via the inhibition of the CCL22–CCR4 axis. Given the crucial role of the CCL22–CCR4 axis in mediating the Treg suppressive function described above (*Rapp et al., 2019*), we propose that the CCL22–CCR4 axis plays a dominant role in the prevention of atherosclerosis by promoting Treg function, which may be inhibited by stimulating the CCL17–CCR4 axis. However, the specific roles of the CCL17–CCR4 and CCL22–CCR4 axes in the various stages of atherosclerosis have not been fully elucidated, and further studies are needed.

This study has several limitations. In flow cytometric analysis, we did not exclude dead cells or doublets. This procedure could have increased the reliability of our data. Although Treg transfer experiments revealed a critical role for CCR4 in Tregs in protecting against early atherosclerosis, the use of conditional knockout mice would provide additional definitive evidence. Our data were obtained from animal experiments, and investigations using human samples will be needed to translate our findings to clinical settings.

In conclusion, we demonstrated that CCR4 protects against early atherosclerosis by favorably modulating the balance between proatherogenic Th1 cell responses and atheroprotective Treg responses. We showed that CCR4 expression on Tregs is critical for suppressing Th1 cell responses and may play an important role in mediating Treg migration to the atherosclerotic aorta. Our data suggest that CCR4 is an important negative regulator of atherosclerosis.

## Methods
### Animals
All mice were male on a C57BL/6 background and fed a standard chow diet or a high-cholesterol diet containing 1.25% cholesterol (CLEA Japan, Tokyo, Japan) as indicated. Wild-type mice were

obtained from CLEA Japan. *Apoe*[-/-11] and *Ccr4*[-/-] mice (*Matsuo et al., 2016*) are previously described. We crossed *Ccr4*[-/-] mice with *Apoe*[-/-] mice to obtain *Ccr4*[-/-]*Apoe*[-/-] mice. Kaede-Tg mice (RBRC05737) (*Tomura et al., 2008*) were provided by the RIKEN BRC through the National BioResource Project of the MEXT/AMED, Japan. We housed mice in cages for each strain or treatment group in a specific pathogen-free animal facility at Kobe Pharmaceutical University. Randomization and allocation concealment were performed. Littermate mice of each genotype were randomly allocated to each experimental group. During the experiments, animal/cage location was not controlled. The investigators were not blinded to the mouse genotype or treatment allocation. The criterion for exclusion was defined as severe body weight loss and set before the study. However, during at least two observations per week, we did not find such symptoms, and no mice were excluded. The experimental procedures were performed in our laboratory rooms or animal facility. All animal experiments were approved by the Animal Care Committee of Kobe Pharmaceutical University (permit numbers: 2018-008, 2019-011, 2020-050, 2021-038, 2022-005, 2023-038, and 2024-014) and conformed to the National Institutes of Health Guide for the Care and Use of Laboratory Animals and the ARRIVE guidelines (Animal Research: Reporting of In Vivo Experiments).

## Assessment of biochemical parameters

Under anesthesia by intraperitoneal injection of medetomidine hydrochloride (0.3 mg/kg), midazolam (4 mg/kg), and butorphanol tartrate (5 mg/kg) (all from WAKO, Osaka, Japan), blood was collected by the cardiac puncture after overnight fasting, and plasma lipid profile was analyzed as described previously (*Tanaka et al., 2024*). Concentrations of plasma total cholesterol, high-density lipoprotein-cholesterol, and triglycerides were determined enzymatically using an automated chemistry analyzer (Oriental Yeast Co., Ltd., Tokyo, Japan).

## Assessment of atherosclerotic lesions

The atherosclerotic lesions in the aortic root and thoracoabdominal aorta were analyzed as described previously (*Tanaka et al., 2024*). For analysis of atherosclerotic lesions in the aortic root, five consecutive sections (10 µm thickness), spanning 600 µm of the aortic sinus (150 µm interval), were collected from each mouse and stained with hematoxylin-eosin. The lesion area of the sections was quantified using ImageJ (National Institutes of Health). Some sections were stained with Oil Red O (Sigma)

**Table 4.** Antibodies for immunohistochemistry.

| Antibodies | Clone | Fluorescent dye | Source |
|---|---|---|---|
| Anti-CCL17 Ab | - | - | abcam |
| | | | ab182793 |
| Anti-MOMA-2 Ab | - | - | BMA Biomedical |
| | | | T-2007 |
| Anti-CCL22 Ab | - | - | R&D |
| | | | AF439 |
| Anti-CD4 Ab | RM4-5 | - | BD Biosciences |
| | | | 550280 |
| Anti-rabbit IgG | - | Alexa Fluor 568 | Thermo Fisher Scientific |
| | | | A11011 |
| Anti-goat IgG | - | Alexa Fluor 568 | Thermo Fisher Scientific |
| | | | A11057 |
| Anti-rat IgG | - | Alexa Fluor 488 | Thermo Fisher Scientific |
| | | | A21208 |
| Anti-rat IgG | - | Biotin | abcam |
| | | | ab102250 |

for representative photomicrographs of the aortic sinus atherosclerotic lesions. For en face analysis of thoracoabdominal aortas, the aorta was opened longitudinally and stained with Oil Red O. The proportion of the lesion area was determined using ImageJ.

## Histological analysis of atherosclerotic lesions

Immunofluorescence staining of CCL17 and CCL22 in lymphoid tissues and atherosclerotic lesions was performed on 4% paraformaldehyde-fixed cryosections of mouse aortic roots using rabbit anti-CCL17 (1:200; abcam) or goat anti-CCL22 (1:200; R&D Systems) antibodies, followed by detection with fluorescent secondary antibodies. For costaining of macrophages in the above cryosections, anti-MOMA-2 (1:400; BMA Biomedicals) and fluorescent secondary antibodies were also used. Stained sections were digitally captured using a fluorescence microscope (BZ-X810; KEYENCE, Osaka, Japan).

For the detection of macrophages or CD4[+] T cells, immunohistochemistry was performed on 4% paraformaldehyde-fixed cryosections of mouse aortic roots using anti-MOMA-2 or anti-CD4 (1:100; BD Biosciences) antibodies, followed by detection with biotinylated secondary antibodies and streptavidin-horseradish peroxidase. Staining with Masson's trichrome (Muto Pure Chemicals, Tokyo, Japan) was used to delineate the fibrous area. Stained sections were digitally captured using a microscope (BZ-X810; KEYENCE), and the percentage of the stained area (the stained area per total atherosclerotic lesion area) was calculated as described previously (*Sasaki et al., 2009*). CD4[+] T cells were quantified as described previously by counting the number of positively stained cells, which was divided by the total plaque area (*Sasaki et al., 2009*). The primary and secondary antibodies used are listed in *Table 4*.

## Flow cytometry

For flow cytometric analysis of lymphoid tissues, peripheral LN cells and splenocytes were isolated and stained in PBS containing 2% fetal calf serum. For analysis of immune cells within the aorta, mice were anesthetized, and the aorta was perfused with cold saline. The aorta was dissected and the adventitial tissue was carefully removed. The aorta was digested with Liberase (Roche Diagnostics) in plain RPMI medium at 37°C for 45 min with vortexing. For the detection of CCR4 on aortic T cells, Collagenase D (Sigma-Aldrich) was used instead of Liberase. A cell suspension obtained by mashing the aorta through a 70 μm strainer was stained with antibodies specific for CD3, CD4, CD45, CCR4, Foxp3, T-bet, RORγt, and GATA3. The Foxp3 staining buffer set (Thermo Fisher Scientific) was used for intracellular staining of Foxp3. In all staining procedures, Fc receptors were blocked by anti-CD16/CD32 (BD Biosciences). Flow cytometric analysis was performed with FACSAria III (BD Biosciences) using FlowJo software version 10.8.1 (Tree Star). The antibodies used were listed in *Table 5*. Gating strategy of flow cytometric analysis of aortic T cells is shown in *Appendix 4—figure 1*.

## Intracellular cytokine staining

Immune cells from lymphoid tissues were stimulated with 20 ng/mL phorbol 12-myristate 13-acetate (Sigma) and 1 mmol/L ionomycin (Sigma) for 5 hours in the presence of Brefeldin A (Thermo Fisher Scientific). Intracellular cytokine staining was performed as described previously (*Tanaka et al., 2024*).

## Cytokine assay

In cell culture experiments, RPMI 1640 medium (Sigma) supplemented with 10% fetal calf serum, 50 μmol/L 2β-mercaptoethanol, and antibiotics was used. The production of several major cytokines from CD4[+] T cells was examined as described previously (*Matsumoto et al., 2016*). Splenic CD4[+] T cells (1×10[5] cells) isolated using MACS (Miltenyi Biotec) were stimulated with plate-bound anti-CD3 (10 μg/mL, clone 145-2C11; BD Biosciences) and soluble anti-CD28 antibodies (2 μg/mL, clone 37.51; BD Biosciences) in 96-well round-bottomed plates for 48 hours. The concentrations of IL-4, IL-10, IL-17, and IFN-γ in culture supernatants were determined by ELISA using paired antibodies specific for corresponding cytokines (R&D Systems). The levels of multiple inflammation-associated cytokines and chemokines in culture supernatants were determined using a Mouse Cytokine Array Kit according to the manufacturer's instructions (R&D Systems).

In some experiments, CD4[+]CD25[+] Tregs and CD4[+]CD25[-] T cells were purified from the peripheral LNs and spleen of *Apoe[-/-]* or *Ccr4[-/-]Apoe[-/-]* mice using a CD4[+]CD25[+] Regulatory T Cell Isolation Kit (Miltenyi Biotec) and anti-CD4 beads (Miltenyi Biotec) according to the manufacturer's instructions.

**Table 5.** Antibodies for flow cytometry.

| Antibodies | Clone | Fluorescent dye | Source |
|---|---|---|---|
| Anti-CD16/CD32 Ab | 2.4G2 | - | BD Biosciences |
| | | | 553142 |
| Anti-CD45 Ab | 30-F11 | PerCPCy5.5 | BD Biosciences |
| | | | 550994 |
| Anti-CD3 Ab | 145-2C11 | | BD Biosciences |
| | | APC | 553066 |
| | | FITC | 553062 |
| | | PECy7 | 552774 |
| Anti-CD4 Ab | RM4-5 | PECy7 | BD Biosciences |
| | | | 552775 |
| Anti-CCR4 Ab | 2G12 | PE | BioLegend |
| | | | 131204 |
| Anti-Foxp3 Ab | FJK-16s | V450 | eBioscience |
| | | | 48-5773-82 |
| Anti-CD44 Ab | IM7 | PE | BD Biosciences |
| | | | 553134 |
| Anti-CD62L Ab | MEL-14 | FITC | BD Biosciences |
| | | | 553150 |
| Anti-Ki67 Ab | SolA15 | FITC | eBioscience |
| | | | 11-5698-82 |
| Anti-CD152 Ab | UC10-4B9 | APC | eBioscience |
| | | | 17-1522-82 |
| Anti-CD103 Ab | M290 | FITC | BD Biosciences |
| | | | 557494 |
| Anti-CD25 Ab | PC61 | PE | BD Biosciences |
| | | | 553866 |
| Anti-CD28 Ab | 37.51 | - | BD Biosciences |
| | | | 553294 |
| Anti-IFNγ Ab | XMG1.2 | PE | BD Biosciences |
| | | | 554412 |
| Anti-IL-4 Ab | 11B11 | PE | BD Biosciences |
| | | | 554435 |
| Anti-IL-10 Ab | JES5-16E3 | APC | BD Biosciences |
| | | | 554468 |
| Anti-IL-17 Ab | TC11-18H10 | APC | BD Biosciences |
| | | | 560184 |
| Anti-T-bet Ab | 4B10 | APC | BioLegend |
| | | | 644814 |
| Anti-Gata3 Ab | L50-823 | Alexa Fluor 488 | BD Biosciences |

*Table 5 continued on next page*

*Table 5 continued*

| Antibodies | Clone | Fluorescent dye | Source |
|---|---|---|---|
| | | | 560163 |
| Anti-RORγt Ab | Q31-378 | PE | BD Biosciences |
| | | | 562607 |
| Anti-CD8 Ab | 53-6.7 | PerCPCy5.5 | BD Biosciences |
| | | | 553033 |
| Anti-B220 Ab | RA3-6B2 | PE | BD Biosciences |
| | | | 553090 |
| Anti-CD11b Ab | M1/70 | V450 | BD Horizon |
| | | | 560455 |
| Anti-Ly6G Ab | 1A8 | FITC | BD Biosciences |
| | | | 551460 |
| Anti-Ly6C Ab | AL-21 | AOC | BD Biosciences |
| | | | 560595 |
| Anti-NK1.1 Ab | PK136 | APC | BD Biosciences |
| | | | 550627 |
| Anti-CD11c Ab | HL3 | V450 | BD Horizon |
| | | | 560521 |
| Anti-I-Ab Ab | AF6-120.1 | FITC | BD Biosciences |
| | | | 553551 |
| Anti-CD80 Ab | 16-10A1 | PE | BD Biosciences |
| | | | 553769 |
| Anti-CD86 Ab | GL1 | APC | BD Biosciences |
| | | | 558703 |
| Anti-CD279 Ab | 29F.1A12 | FITC | BioLegend |
| | | | 135213 |

The purity of each population was >95% and most of the CD4$^+$CD25$^+$ T cells expressed Foxp3, as determined by flow cytometric analysis. The viability of each cell population was analyzed using 7-amino-actinomycin D (BD Biosciences) and Alexa Fluor 488 Annexin V/Dead Cell Apoptosis Kit (Thermo Fisher Scientific). The detailed data are shown in *Appendix 5—figure 1A–D*. CD11c$^+$ DCs were isolated from spleen of *Apoe*$^{-/-}$ mice treated with Liberase using MACS (Miltenyi Biotec). The purity of the CD11c$^+$ population was approximately 93%, as determined by flow cytometric analysis. CD4$^+$CD25$^-$ T cells (1×10$^5$ cells) from *Apoe*$^{-/-}$ mice and splenic CD11c$^+$ DC (2×10$^4$ cells) were cocultured with or without CD4$^+$CD25$^+$ Tregs (1.25×10$^4$ cells) from *Apoe*$^{-/-}$ or *Ccr4*$^{-/-}$*Apoe*$^{-/-}$ mice in the presence of soluble anti-CD3 antibody (0.5 μg/mL) in 96-well round-bottomed plates. Culture supernatants were collected at 48 hours and analyzed by ELISA for IFN-γ as described above.

## Treg suppression assay

For analysis of the in vitro suppressive function of Tregs, CD4$^+$CD25$^+$ Tregs and CD4$^+$CD25$^-$ T cells were purified from pooled peripheral LNs and spleen of *Apoe*$^{-/-}$ or *Ccr4*$^{-/-}$*Apoe*$^{-/-}$ mice as described above. Purified CD4$^+$CD25$^+$ Tregs from *Apoe*$^{-/-}$ or *Ccr4*$^{-/-}$*Apoe*$^{-/-}$ mice were cocultured with carboxy-fluorescein diacetate succinimidyl ester (Thermo Fisher Scientific)-labeled CD4$^+$CD25$^-$ conventional T cells (2.5×10$^4$ cells) from *Apoe*$^{-/-}$ mice at the indicated ratios in the presence of mitomycin C (WAKO)-treated antigen-presenting cells (5×10$^4$ cells) and soluble anti-CD3 antibody (0.5 μg/mL) in 96-well

round-bottomed plates. Splenocytes were used as antigen-presenting cells. The cocultured cells were maintained at 37°C with 5% $CO_2$ for 3 days. The proliferation of carboxyfluorescein diacetate succinimidyl ester-labeled CD4+CD25- conventional T cells was analyzed by flow cytometry.

## Flow cytometric analysis of DC phenotypic changes

CD4+CD25+ Tregs and CD4+CD25- T cells were purified from peripheral LNs and spleen of Apoe-/- or Ccr4-/-Apoe-/- mice, and CD11c+ DCs were isolated from the spleen of Apoe-/- mice as described above. CD4+CD25- T cells (5×10^4 cell) from Apoe-/- mice or a mixture of CD4+CD25+ Tregs (5×10^4 cells) from Apoe-/- or Ccr4-/-Apoe-/- mice and CD4+CD25- T cells (5×10^4 cells) from Apoe-/- mice were cocultured with splenic CD11c+ DCs (4×10^4 cells) in the presence of soluble anti-CD3 antibody (0.1 μg/mL) in 96-well round-bottomed plates. After 48 hours of coculture, the cells were collected, stained with antibodies specific for IAb, CD11c, CD80, CD86, and 7-amino-actinomycin D, and analyzed using FACSAria III (BD Biosciences).

## Quantitative reverse transcription PCR analysis

Using TRIzol reagent (Thermo Fisher Scientific), we extracted total RNA from aorta which was perfused with cold saline and subsequently soaked in RNA later (Thermo Fisher Scientific). After the isolation of CD4+CD25+ Tregs and CD4+CD25- T cells as described above, we extracted total RNA from the cells using an RNeasy Mini Kit (QIAGEN). A PrimeScript RT reagent Kit (Takara, Shiga, Japan) was used

**Table 6.** Primer sequences for quantitative reverse transcription PCR.

| Gene | Forward primer sequence (5'→3') | Reverse primer sequence (5'→3') |
|---|---|---|
| Gapdh | TGTGTCCGTCGTGGATCTGA | TTGCTGTTGAAGTCGCAGGAG |
| Il1b | TCCAGGATGAGGACATGAGCAC | GAACGTCACACACCAGCAGGTTA |
| Il6 | CCACTTCACAAGTCGGAGGCTTA | GCAAGTGCATCATCGTTGTTCATAC |
| Il10 | GACCAGCTGGACAACATACTGCTAA | GATAAGGCTTGGCAACCCAAGTAA |
| Tnf | CCACCACGCTCTTCTGTCTAC | AGGGTCTGGGCCATAGAACT |
| Ifng | CGGCACAGTCATTGAAAGCCTA | GTTGCTGATGGCCTGATTGTC |
| Tbx21 | CTGCCTACCAGAACGCAGA | AAACGGCTGGGAACAGGA |
| Gata3 | GGATGTAAGTCGAGGCCCAAG | ATTGCAAAGGTAGTGCCCGGTA |
| Rorc | CACAGAGACACCACCGGACAT | CGTGCAGGAGTAGGCCACATT |
| Foxp3 | CTCATGATAGTGCCTGTGTCCTCAA | AGGGCCAGCATAGGTGCAAG |
| Ctla4 | CCTCTGCAAGGTGGAACTCATGTA | AGCTAACTGCGACAAGGATCCAA |
| Cd103 | ATGGCATTCAGTGGTCTGTGCTA | CACCAAGGATCGGCAGTTCA |
| Tnfrsf18 | GTTCAGAACGGAAGTGGCAACA | GCTTGCAGATCTTGCACTGAGG |
| Tgfb | GTGTGGAGCAACATGTGGAACTCTA | TTGGTTCAGCCACTGCCGTA |
| Cd44 | CTGGCACTGGCTCTGATTCTTG | TCCCATTGCCACCGTTGA |
| Cd69 | TGGCCCAACGCTCTTGTTC | GCCCAATCCAATGTTCCAGTTC |
| Ccr4 | TCTACAGCGGCATCTTCTTCAT | CAGTACGTGTGGTTGTGCTCTG |
| Ccr5 | CCTAGCCAGAGGAGGTGAGACATC | AGCTATAGGTCGGAACTGACCCTTG |
| Ccr6 | GGCAGTTACTCATGCCACCAA | GGAGCAGCATCCCACAGTTAAAG |
| Ccr7 | GGTGGTGGCTCTCCTTGTCATT | ACACCGACTCGTACAGGGTGTAGTC |
| Ccr8 | CAGACCCACAACCTGCTGGA | GACAGCGTGGACAATAGCCAGA |
| Ccr1 | GGTTGGGACCTTGAACCTTG | GGGTAGGCTTCTGTGAAATCTG |
| Cxcr3 | ATCACCTGGTGGTGCTAGTGGA | AAAGGCATAGAGCAGCGGATTG |
| Cx3cr1 | AAGCACTTGCCTCTGGTGGA | AGGCCTCAGCAGAATCGTCATA |

for reverse transcription. Quantitative PCR analysis was conducted using a TB Green Ex Taq (Takara) and a StepOnePlus Real-Time PCR System (Thermo Fisher Scientific) according to the manufacturer's instructions. The primers used are listed in *Table 6*. Amplification reactions were performed in duplicate, and fluorescence curves were analyzed with the included software. GAPDH was used as an endogenous control reference.

### In vivo Treg homing assay

CD4+CD25+ Tregs were purified from the peripheral LNs and spleen of Apoe-/-Kaede-Tg or Ccr4-/-Apoe-/-Kaede-Tg mice as described above and were intravenously injected into recipient 18-week-old Apoe-/- mice (1×10⁶ cells per mouse) on a high-cholesterol diet containing 1.25% cholesterol via the tail vein. At 20 hours after transfer, the Kaede+ Treg proportions in the peripheral LNs, spleen, para-aortic LNs, and aorta of Apoe-/- mice were analyzed by flow cytometry. Gating strategy of flow cytometric analysis of Kaede-expressing Tregs in peripheral lymphoid tissues and aortas was shown in *Appendix 6—figure 1*.

### Analysis of atherosclerotic lesions in Treg-transferred mice

To clarify the role of CCR4 expression in Tregs in regulating atherosclerosis, CD4+CD25+ Tregs were purified from the peripheral LNs and spleen of *Apoe-/-* or *Ccr4-/-Apoe-/-* mice as described above and were intravenously injected into 12-week-old recipient *Apoe-/-* or *Ccr4-/-Apoe-/-* mice (1×10⁶ cells per mouse) on a standard chow diet via the tail vein. The atherosclerotic lesions were analyzed 4 weeks after transfer as described above.

### Statistical analysis

Normality was assessed using the Shapiro–Wilk normality test. Two-tailed Student's *t*-test, Mann–Whitney *U*-test, or one-sample *t*-test was used to detect significant differences between the two groups when appropriate. One-way ANOVA followed by Tukey's multiple-comparisons test or two-way ANOVA followed by Tukey's multiple-comparisons test was performed for multiple groups where appropriate. A value of p<0.05 was considered statistically significant. No data were excluded from the analysis. The investigators were not blinded to the data analysis. For statistical analysis, GraphPad Prism version 9.0 (GraphPad Software Inc) was used.

## Acknowledgements

Kaede-Tg mice (RBRC05737) were provided by the RIKEN BRC through the National BioResource Project of the MEXT/AMED, Japan. This work was supported by Japan Society for the Promotion of Science (JSPS) KAKENHI Grant Numbers JP21K08042 and JP24K11301 (to NS) and research grants from Pfizer Japan Inc (to YR), Astellas Pharma Inc (to YR), Novartis Pharma KK (to NS), Senshin Medical Research Foundation (to NS), and ONO Medical Research Foundation (to NS).

## Additional information

### Funding

| Funder | Grant reference number | Author |
|---|---|---|
| KAKENHI | JP21K08042 | Naoto Sasaki |
| SENSHIN Medical Research Foundation | | Naoto Sasaki |
| Ono Medical Research Foundation | | Naoto Sasaki |
| Novartis Japan | | Naoto Sasaki |
| Pfizer Japan | | Yoshiyuki Rikitake |
| Astellas Pharma | | Yoshiyuki Rikitake |

| Funder | Grant reference number | Author |
|---|---|---|
| KAKENHI | JP24K11301 | Naoto Sasaki |

The funders had no role in study design, data collection and interpretation, or the decision to submit the work for publication.

## Author contributions

Toru Tanaka, Data curation, Formal analysis, Validation, Investigation, Visualization, Methodology, Writing – original draft, Project administration, Writing – review and editing; Naoto Sasaki, Conceptualization, Resources, Data curation, Formal analysis, Supervision, Funding acquisition, Validation, Investigation, Visualization, Methodology, Writing – original draft, Project administration, Writing – review and editing; Aga Krisnanda, Hilman Zulkifli Amin, Ken Ito, Validation, Investigation, Writing – review and editing; Sayo Horibe, Ken-ichi Hirata, Writing – review and editing; Kazuhiko Matsuo, Takashi Nakayama, Yoshiyuki Rikitake, Resources, Writing – review and editing

## Author ORCIDs

Toru Tanaka (ORCID) https://orcid.org/0000-0002-1054-9211
Naoto Sasaki (ORCID) https://orcid.org/0000-0002-7760-6129
Aga Krisnanda (ORCID) https://orcid.org/0009-0005-6417-2738
Sayo Horibe (ORCID) https://orcid.org/0000-0002-4111-2426
Kazuhiko Matsuo (ORCID) https://orcid.org/0000-0001-5782-5300
Takashi Nakayama (ORCID) https://orcid.org/0000-0002-8493-899X
Yoshiyuki Rikitake (ORCID) https://orcid.org/0000-0001-7207-4656

## Ethics

The experimental procedures were performed in our laboratory rooms or animal facility. All animal experiments were approved by the Animal Care Committee of Kobe Pharmaceutical University (permit numbers: 2018-008, 2019-011, 2020-050, 2021-038, 2022-005, 2023-038, and 2024-014) and conformed to the National Institutes of Health Guide for the Care and Use of Laboratory Animals and the ARRIVE guidelines (Animal Research: Reporting of In Vivo Experiments).

Reviewer #2 (Public review): https://doi.org/10.7554/eLife.101830.4.sa1
Reviewer #3 (Public review): https://doi.org/10.7554/eLife.101830.4.sa2
Author response https://doi.org/10.7554/eLife.101830.4.sa3

# Additional files

## Supplementary files

MDAR checklist

## Data availability

All data generated or analyzed during this study are included in the manuscript, appendix figures, and figure supplements.

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

## Appendix 1

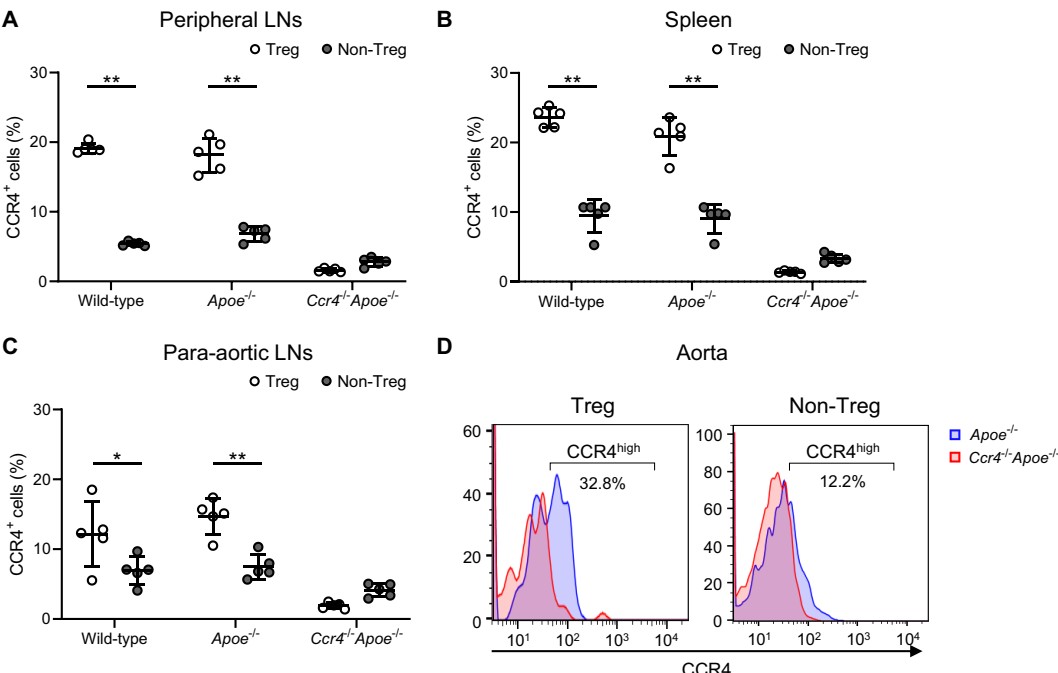

**Appendix 1—figure 1.** C-C chemokine receptor 4 (CCR4) is predominantly expressed on CD4+Foxp3+ Tregs. (**A–C**), The graphs represent the proportions of CCR4+ cells in CD4+Foxp3+ Tregs and CD4+Foxp3- non-Tregs in the peripheral lymph nodes (LNs) (**A**), spleen (**B**), and para-aortic LNs (**C**) of 18-week-old wild-type, *Apoe-/-*, or *Ccr4-/-Apoe-/-* mice assessed by flow cytometry. n=5 per group. Data points represent individual animals. Horizontal bars represent means. Error bars indicate s.d. (**D**) Representative flow cytometric analysis of CCR4 expression in aortic CD4+Foxp3+ Tregs and CD4+Foxp3- non-Tregs from 18-week-old *Apoe-/-* or *Ccr4-/-Apoe-/-* mice. Pooled aortic lymphoid cells from 7 to 8 mice in each group were used. Data are representative of two independent experiments. *p<0.05, **p<0.01; two-way ANOVA followed by Tukey's multiple-comparisons test: (**A**, **B** and **C**).

The online version of this article includes the following source data for appendix 1—figure 1:

**Appendix 1—figure 1—source data 1.** Raw numerical values for *Appendix 1—figure 1* plots.

## Appendix 2

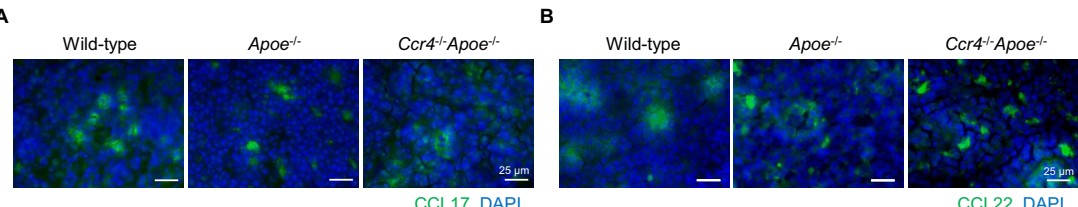

**Appendix 2—figure 1.** CCL17 and CCL22 are detected in peripheral lymph nodes (LNs). Immunostaining for CCL17 (green) (**A**) and CCL22 (green) (**B**) in the peripheral LNs of 18-week-old wild-type, *Apoe*-/-, or *Ccr4*-/-*Apoe*-/- mice. Nuclei were stained with DAPI (blue). Data are representative of five mice analyzed in each group. White bars represent 25 μm as described.

## Appendix 3

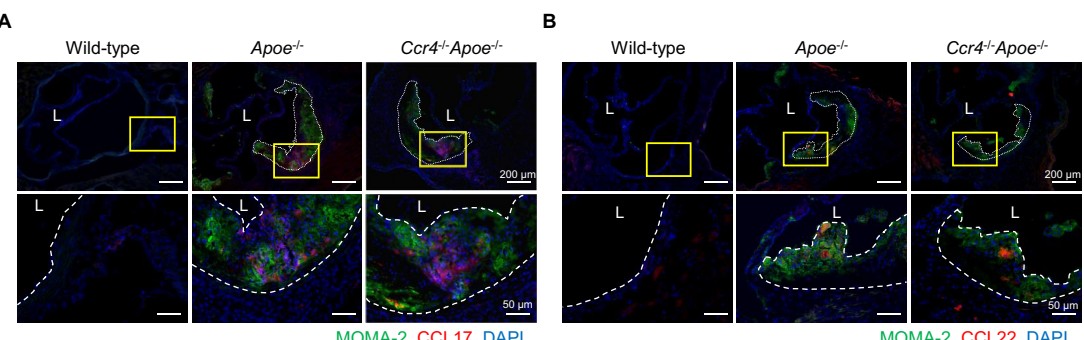

**Appendix 3—figure 1.** CCL17 and CCL22 are detected in atherosclerotic lesions. Immunostaining for CCL17 (red) and MOMA-2 (green) (**A**) and for CCL22 (red) and MOMA-2 (green) (**B**) in the aortic sinus of 18-week-old wild-type, *Apoe*-/-, or *Ccr4*-/-*Apoe*-/- mice. Boxed area is expanded to show high-power fields. Nuclei were stained with DAPI (blue). Dashed lines demarcate atherosclerotic lesions or indicate the inner lining of arteries; L, lumen. Data are representative of five mice analyzed in each group. White bars represent 50 or 200 µm as described.

## Appendix 4

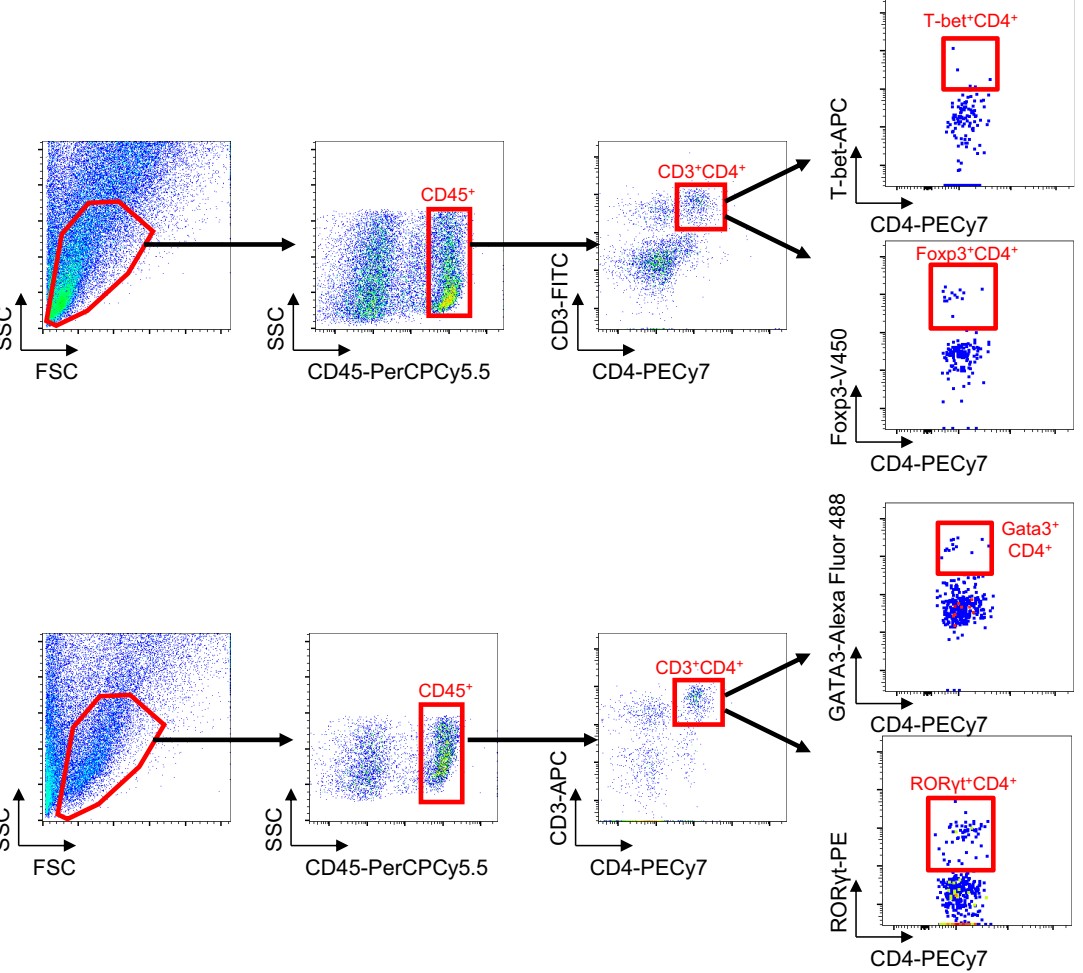

**Appendix 4—figure 1.** Gating strategy of flow cytometric analysis of T-bet, GATA3, RORγt, and Foxp3 expression in aortic CD3$^+$CD4$^+$CD45$^+$ T cells.

## Appendix 5

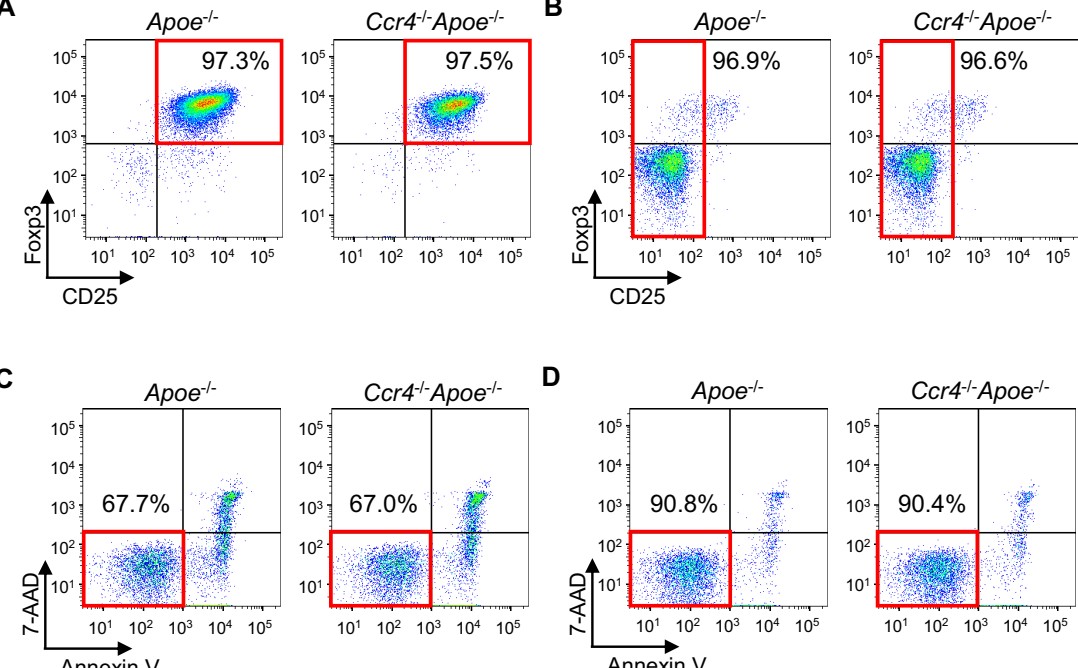

**Appendix 5—figure 1.** The purity and viability of CD4+CD25+ Tregs and CD4+CD25 T cells isolated from *Apoe-/-* or *Ccr4-/-Apoe-/-* mice. Representative flow cytometric analysis of CD25 and Foxp3 expression in CD4+CD25+ Tregs (**A**) and CD4+CD25- T cells (**B**) purified from peripheral lymphoid tissues of *Apoe-/-* or *Ccr4-/-Apoe-/-* mice. Representative flow cytometric analysis of the viability of CD4+CD25+ Tregs (**C**) and CD4+CD25- T cells (**D**) purified from peripheral lymphoid tissues of *Apoe-/-* or *Ccr4-/-Apoe-/-* mice. T cells which neither expressed 7-AAD nor Annexin V were considered viable.

## Appendix 6

Peripheral lymphoid tissues

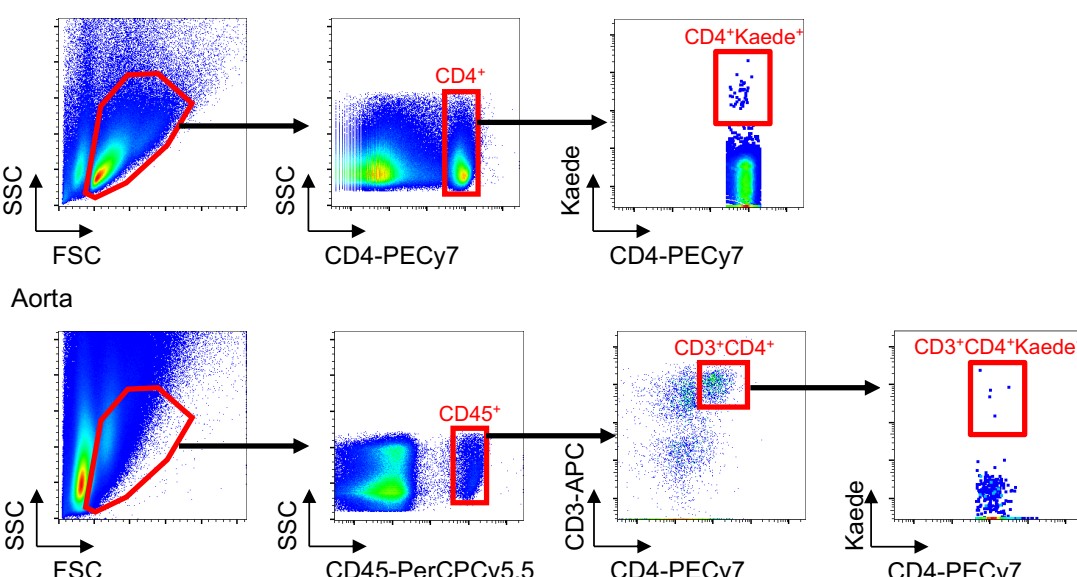

**Appendix 6—figure 1.** Gating strategy of flow cytometric analysis of Kaede-expressing Tregs in peripheral lymphoid tissues and aortas.

