## [Editor Report · eLife Assessment]

This **valuable** study provides in vivo evidence that CCR4 regulates the early inflammatory response during atherosclerotic plaque formation. The authors propose that altered T-cell response plays a role in this process, shedding light on mechanisms that may be of interest to medical biologists, biochemists, cell biologists, and immunologists. The work is currently considered **incomplete** pending textual changes and the inclusion of proper controls.

---

## [Referee Report · Reviewer #2 (Public review)]

Summary:

Tanaka et al. investigated the role of CCR4 in early atherosclerosis, focusing on the immune modulation elicited by this chemokine receptor under hypercholesterolemia. The study found that Ccr4 deficiency led to qualitative changes in atherosclerotic plaques, characterized by an increased inflammatory phenotype. The authors further analyzed the CD4 T cell immune response in para-aortic lymph nodes and atherosclerotic aorta, showing an increase mainly in Th1 cells and the Th1/Treg ratio in Ccr4-/-Apoe-/- mice compared to Apoe-/- mice. They then focused on Tregs, demonstrating that Ccr4 deficiency impaired their immunosuppressive function in in vitro assays. Authors also states that Ccr4-deficient Tregs had, as expected, impaired migration to the atherosclerotic aorta. Adoptive cell transfer of Ccr4-/- Tregs to Apoe-/- mice mimicked early atherosclerosis development in Ccr4-/-Apoe-/- mice. Therefore, this work shows that CCR4 plays an important role in early atherosclerosis but not in advanced stages.

Strengths:

Several in vivo and in vitro approaches were used to address the role of CCR4 in early atherosclerosis. Particularly, through the adoptive cell transfer of CCR4+ or CCR4- Tregs, the authors aimed to demonstrate the role of CCR4 in Tregs' protection against early atherosclerosis.

Weaknesses:

Flow cytometry experiments are not well controlled. Dead cells and doublets were not excluded from analysis.

Clinical relevance is unclear.

Comments on revisions:

I thank the authors for addressing my suggestions.

I understand that excluding dead cells would require repeating the entire experiment. However, the authors can at least exclude doublets from the existing flow cytometry data.

I also agree with the more cautious claim regarding the role of CCR4 in Treg migration.

---

## [Referee Report · Reviewer #3 (Public review)]

Summary

Tanaka and colleagues addressed the role of the C-C chemokine receptor 4 (CCR4) in early atherosclerotic plaque development using ApoE-deficient mice on a standard chow diet as a model. Because several CD4+ T cell subsets express CCR4, they examined whether CCR4-deficiency alters the immune response mediated by CD4+ T cells. By histological analysis of aortic lesions, they demonstrated that the absence of CCR4 promoted the development of early atherosclerosis, with heightened inflammation linked to increased macrophages and pro-inflammatory CD4+ T cells, along with reduced collagen content. Flow cytometry and mRNA expression analysis for identifying CD4+ T cell subsets showed that CCR4 deficiency promoted higher proliferation of pro-inflammatory effector CD4+ T cells in peripheral lymphoid tissues and accumulation of Th1 cells in the atherosclerotic lesions. Interestingly, the increased pro-inflammatory CD4+ T cell response occurred despite the expansion of T CD4+ Foxp3+ regulatory cells (Tregs), found in higher numbers in lymphoid tissues of CCR4-deficient mice, suggesting that CCR4 deficiency interfered with Treg's regulatory actions. The findings contrast with earlier studies in a murine model of advanced atherosclerosis, where CCR4 deficiency did not alter the development of the aortic lesions. The authors included a thoughtful discussion about hypothetical mechanisms explaining these contrasting results, including putative differences in the role played by the CCL17/CCL22-CCR4 axis along the stages of atherosclerosis development in this murine model.

Major strengths

• Demonstration of CCR4 deficiency's impact on early atherosclerosis. CCR4 deficiency effects on the early atherosclerosis development in the Apoe-/-mice model were demonstrated by a quantitative analysis of the lesion area, inflammatory cell content and the expression profile of several pro- and anti-inflammatory markers.

• Analysis of the T CD4+ response in various lymphoid tissues (peripheral and para-aortic lymph nodes and spleen) and the atherosclerotic aorta during the early phase of atherosclerosis in the Apoe-/-mice model. This analysis, combining flow cytometry and mRNA expression, showed that CCR4 deficiency enhanced T CD4+ cell activation, favouring the amplification of the typical biased Th1-mediated inflammatory response observed in the lymphoid tissues of hypercholesterolemic mice.

• Treg transference experiments. Transference of Treg from Apoe-/- or Ccr4-/- Apoe-/- mice to Apoe-/- mice under a standard chow diet was useful for addressing the relevance of CCR4 expression on Tregs for the atheroprotective effect of this regulatory T cell subset during early atherosclerosis.

Major weaknesses

• Methodological Limitations: The controls used in the flow cytometry analysis were suboptimal, as neither cell viability nor doublets were assessed. This may have introduced artifacts, particularly when measuring less-represented cell populations within complex samples, such as in assays evaluating Treg migration to the aorta in atherosclerotic mice.

• Incomplete understanding of CCR4-Mediated Mechanisms: The mechanisms by which CCR4 regulates early inflammation and the development of atherosclerosis were not fully clarified.

I have previously addressed the study limitations and their global impact in my earlier reviews.

---

## [Author Response]

The following is the authors’ response to the previous reviews

**Response to the reviewer #2 (Public review):**

We greatly appreciate the reviewer’s high evaluation of our paper and helpful comments and suggestions.

Regarding in vivo Treg homing assay, we did not exclude doublets and dead cells from the analysis of Kaede-expressing Tregs migrated to the aorta, which may affect the results. We described this issue as the limitation of this study in the revised manuscript. Nonetheless, we believe the reliability of our findings because we repeated this experiment three times and obtained similar results.

There is no evidence to support the clinical relevance of our findings. Future clinical research on this topic is highly desired.

**Response to the reviewer #3 (Public review):**

We greatly appreciate the reviewer’s high evaluation of our paper and helpful comments and suggestions.

Despite the controversial role of Th17 cells in atherosclerosis, we understand the possible involvement of Th17 cells and the Th1 cell/Th17 cell balance in lymphoid tissues and aortic lesions in accelerated inflammation and atherosclerosis in *Ccr4*^-/-^*Apoe*^-/-^ mice. Although we could not completely evaluate the changes in these immune responses in detail, future study may elucidate interesting mechanisms mediated by Th17 cell responses.

As the reviewer suggested, we understand that it is necessary to provide in vivo evidence for the Treg suppressive effects on DC activation. Based on the results of in vitro experiments, we described the discussion on the in vivo evidence in the revised manuscript.

We understand methodological limitations for flow cytometric analysis of immune cells in the aorta and in vivo Treg homing assay. We described this issue as the limitation of this study in the revised manuscript. Regarding in vivo Treg homing assay, we statistically re-analyzed the combined data from multiple experiments and observed a tendency toward reduction in the proportion of CCR4-deficient Kaede-expressing Tregs in the aorta of recipient *Apoe*^-/-^ mice, though there was no statistically significant difference in the migratory capacity of CCR4-intact or CCR4-deficient Kaede-expressing Tregs. Accordingly, we toned down our claim that CCR4 expression on Tregs plays a critical role in mediating Treg migration to the atherosclerotic aorta under hypercholesterolemia.

The reviewer requested us to evaluate aortic inflammation in *Ccr4*^-/-^*Apoe*^-/-^ mice injected with CCR4-intact or CCR4-deficient Tregs. However, we think that this experiment will provide marginal information because Treg transfer experiments in *Apoe*^-/-^ mice have already shown the protective role of CCR4 in Tregs against aortic inflammation and early atherosclerosis.

**Recommendations for the authors:**

**Reviewer #2 (Recommendations for the authors):**
(1) #1 and #2: CD103 and CD86 expression should be discussed on the text and not only in the response to reviewer.

In accordance with the reviewer’s suggestion, we added a discussion on the downregulated CD103 expression in peripheral LN Tregs and upregulated CD86 expression on DCs in *Ccr4*^-/-^*Apoe*^-/-^ mice in the discussion section in the revised manuscript.

(2) #5: Authors response is not satisfactory. No gate percentage is shown. As it currently is, the difference in the number of cells shown in the figure could be due to differences in events recorded. Furthermore, the gate strategy is not thorough. Considering the very low frequency of Kaede + cells detected, it is crucial to properly exclude doublets and dead cells.Authors reported a dramatic difference in Kaede + Tregs cells in the aorta across experiments. This could be addressed by normalization followed by appropriate statistical analysis (One sample t-test).The data shown is not strong enough to conclude that there is a reduced migration to the aorta.

We understand the importance of reviewer’s suggestion. We described the percentage of Kaede+ Tregs in the aorta of *Apoe*^-/-^ mice receiving transfer of Kaede-expressing CCR4-intact or CCR4-deficient Tregs in Figure 5I.

As the reviewer pointed out, we understand that it would be important to properly exclude doublets and dead cells in in vivo Treg homing assay. However, it is difficult for us to resolve this issue because we need to perform the same experiments again which will require a great number of additional mice and substantial amount of time. We deeply regret that these important experimental procedures were not performed. We described this issue as the limitation of this study.

In accordance with the reviewer’s suggestion, we re-analyzed the combined data from multiple experiments using one-sample *t*-test. We observed a tendency toward reduction in the proportion of CCR4-deficient Kaede-expressing Tregs in the aorta of recipient *Apoe*^-/-^ mice, though there was no statistically significant difference in the migratory capacity of CCR4-intact or CCR4-deficient Kaede-expressing Tregs. By modifying the corresponding descriptions in the manuscript, we toned down our claim that CCR4 expression on Tregs plays a critical role in mediating Treg migration to the atherosclerotic aorta under hypercholesterolemia.

(3) #8: There are still several not shown data

In accordance with the reviewer’s suggestion, we showed the data on the responses of Tregs and effector memory T cells in 8-week-old wild-type or *Ccr4*^-/-^ mice and *Ccr4* mRNA expression in Tregs and non-Tregs from *Apoe*^-/-^ or *Ccr4*^-/-^*Apoe*^-/-^ mice in Supplementary Figures 4 and 7.

**Reviewer #3 (Recommendations for the authors):**

(1) Issue 1. For future studies, I recommend not omitting viability controls during cell staining. Removal of dead cells and doublets should always be included during the gating strategy to avoid undesirable artefacts, especially when analysing less-represented cell populations. According to your previous report (ref #40), I agree that isotype controls were unnecessary using the same staining protocol. FMO controls should always be included in flow cytometry analysis (not mentioned in the methodology description and ref#40).

As the reviewer suggested, we understand that it would be important to properly exclude dead cells and doublets and to prepare FMO controls in flow cytometric analysis. We deeply regret that these important experimental procedures were not performed. We described this issue as the limitation of this study.

(2) Issue 3. Although Th17's role in atherosclerosis remains controversial, the data obtained in this work could provide valuable insights if discussed appropriately. As noted in my public review, I found it noteworthy that ROR γ *t*+ cells represented around 13% of effector TCD45+CD3+CD4+ lymphocytes in the aorta of *Apoe*^-/-^ mice while Th1 less than 5% (Fig 4H and F, respectively). I recognise that differences in cell staining sensibility and robustness for different transcription factors may influence these percentages. However, analysing how CCR4 deficiency influences the Th1/TI h17 balance would yield interesting data, similar to what was done for the Th1/Treg ratio.

Considering the higher proportion of Th17 cells than Th1 or Th2 cells in atherosclerotic aorta, we understand the importance of reviewer’s suggestion. However, we could not evaluate the effect of CCR4 deficiency on the Th1/Th17 balance in aorta because we did not perform flow cytometric analysis of aortic Th1 and Th17 cells in the same mice. Meanwhile, we could examine the Th1/Th17 balance in peripheral lymphoid tissues by flow cytometry. We found a significant increase in the Th1/Th17 ratio in the peripheral LNs of *Ccr4*^-/-^*Apoe*^-/-^ mice, while there were no changes in its ratio in the spleen or para-aortic LNs of these mice, which limits the contribution of the Th1/Th17 balance to exacerbated atherosclerosis. We showed these data below.

(3) Issue 4. I appreciate the authors for sharing data on the flow cytometry analysis of Tregs in para-aortic LNs of *Apoe*^-/-^ and *Ccr4*^-/-^
*Apoe*^-/-^ mice, which would have been included as a Supplementary figure. These results reinforce the notion that Treg dysfunction in CCR4-deficient mice may not be due to the downregulation of regulatory cell surface receptors.

We showed the data on the expression of CTLA-4, CD103, and PD1 in Tregs in the para-aortic LNs of *Apoe*^-/-^ and *Ccr4*^-/-^*Apoe*^-/-^ mice in Supplementary Figure 8.

(4) Issue 5. I agree that CD4+ T cell responses are substantially regulated by DCs. While CD80 and CD86 on DC primarily serve as costimulatory signals for T-cell activation, cytokines secreted by DCs are primordial signals for determining the differentiation phenotype of effector Th cells. Since the analysis of DC phenotype in lymphoid tissues of *Apoe*^-/-^ and *Ccr4*^-/-^
*Apoe*^-/-^ mice could not be addressed in this study, it is not possible to differentiate which processes may be mainly affected by CCR4-deficiency during CD4+ T cell activation. In this scenario, and considering in vitro studies, the results suggest a possible role of CCR4 in controlling the extent of activation of CD4+T cells rather than shifting the CD4+T cell differentiation profile in peripheral lymphoid tissues, where a predominant Th1 profile was already established in *Apoe*^-/-^ mice. Therefore, I advise caution when concluding about shifts in CD4+ T cell responses.

We thank the reviewer for providing us thoughtful comments. As the reviewer pointed out, we understand that we should carefully interpret the mechanisms for the shift of CD4+ T cell responses by CCR4 deficiency.

(5) Regarding migration studies in the revised manuscript. I fully understand that Treg transference assays are challenging. The results do not suggest that CCR4 was critical for Treg migration to lymphoid tissues in the conditions assayed. Concerning migration to the aorta, I found the results inconclusive since the authors mention that: (i) there was a dramatic difference in the absolute numbers of Kaede-expressing Tregs that migrated to the aorta impairing statistical analysis; (ii) the number of Kaede-expressing Tregs that migrated to the aorta was extremely low; (iii) dead cells and doublets were not removed in the flow cytometry analysis. In this context, I do not agree with the following statements and recommend revising them:- "CCR4 deficiency in Tregs impaired their migration to the atherosclerotic aorta" (lines 36-7),- "…we found a significant reduction in the proportion of CCR4 deficient Kaede-expressing Tregs in the aorta of recipient *Apoe*^-/-^ mice" (lines 356-7),- "CCR4 expression on Tregs regulates the development of early atherosclerosis by....... mediating Treg migration to the atherosclerotic aorta" (lines 409-411),- "…we found that CCR4 expression on Tregs is critical for regulating atherosclerosis by mediating their migration to the atherosclerotic aorta" (lines 437-438),- "CCR4 protects against early atherosclerosis by mediating Treg migration to the aorta.... (lines 464-465),- "We showed that CCR4 expression on Tregs is critical for ...... mediating Treg migration to the atherosclerotic aorta" (503-505).

We understand the importance of the reviewer’s suggestion. We described this issue as the limitation of this study. In accordance with the reviewer’s suggestion, we modified the above descriptions and toned down our claim that CCR4 expression on Tregs plays a critical role in mediating Treg migration to the atherosclerotic aorta under hypercholesterolemia.

(6) Line 206: Mention the increased expression of CD86 by DCs

We mentioned this result in the revised manuscript. We also added a discussion on the upregulated CD86 expression on DCs in *Ccr4*^-/-^*Apoe*^-/-^ mice in the discussion section in the revised manuscript.

(7) Lines 304-305. According to Fig 4F-H, a selective accumulation of Th1 cells seems to have occurred only in the aorta, coinciding with a higher Th1/Treg ratio. No selective accumulation of Th1 cells was observed in para-aortic lymph nodes. These results could be clarified.

We modified the above description in the revised manuscript.